# Asymmetric Ru-In atomic pairs promote highly active and stable acetylene hydrochlorination

Yurui Fan[1,5], Haomiao Xu [1,5]✉, Guanqun Gao[1], Mingming Wang[1], Wenjun Huang[1], Lei Ma[1], Yancai Yao[1]✉, Zan Qu [1]✉, Pengfei Xie [2]✉, Bin Dai[3] & Naiqiang Yan [1,4]

Ru single-atom catalysts have great potential to replace toxic mercuric chloride in acetylene hydrochlorination. However, long-term catalytic stability remains a grand challenge due to the aggregation of Ru atoms caused by over-chlorination. Herein, we synthesize an asymmetric Ru-In atomic pair with vinyl chloride monomer yield (>99.5%) and stability (>600 h) at a gas hourly space velocity of 180 h$^{-1}$, far surpassing those of the Ru single-atom counterparts. A combination of experimental and theoretical techniques reveals that there is a strong *d-p* orbital interaction between Ru and In atoms, which not only enables the selective adsorption of acetylene and hydrogen chloride at different atomic sites but also optimizes the electron configuration of Ru. As a result, the intrinsic energy barrier for vinyl chloride generation is lowered, and the thermodynamics of the chlorination process at the Ru site is switched from exothermal to endothermal due to the change of orbital couplings. This work provides a strategy to prevent the deactivation and depletion of active Ru centers during acetylene hydrochlorination.

Acetylene ($C_2H_2$) hydrochlorination is a critical industrial process for the production of vinyl chloride monomer (VCM), which accounts for 35% of global poly(vinyl chloride) (PVC) output. The conventional catalysts, activated carbon-supported mercuric chloride ($HgCl_2$/AC), have faced strong restrictions due to the Minamata Convention on Mercury, thus demanding green, Hg-free alternatives such as noble metal catalysts (e.g., Au, Pd, Ru, etc.)[1–5]. In recent years, ruthenium single-atom catalysts (Ru SACs) have emerged as promising candidates due to their excellent chlorine affinity and flexible control of active site architectures[6–10]. However, Ru SACs always suffer from easy deactivation due to metal aggregation induced by over-chlorination and simultaneous coke deposition (Fig. 1a)[11–15]. In addition, the steric hindrance of the transition state during the reaction at the atomic site substantially impedes the further improvement of activity[16–19]. Therefore, the rational design of Ru SACs with high activity and stable performance towards acetylene hydrochlorination remains a grand challenge.

Recently, the synthesis efforts have been expanded to neighboring sites of single-atom metal that give rise to the bifunctional site at the atomic level and unlock excellent performance for acetylene hydrochlorination[20–24]. Inspired by these previous studies, the fabrication of dual-atom catalysts offers an effective strategy to enhance the reactivity and stability as a result of optimizing the activation behavior of acetylene and hydrogen chloride (HCl) through orbital coupling between two adjacent metal atoms (e.g., *d–p* hybridization)[25–30]. In addition, the *p* block element indium (In) is known to have abundant

[1]School of Environmental Science and Engineering, Shanghai Jiao Tong University, 200240 Shanghai, China. [2]College of Chemical and Biological Engineering, Zhejiang University, 310058 Hangzhou, China. [3]State Key Laboratory Incubation Base for Green Processing of Chemical Engineering, School of Chemistry and Chemical Engineering, Shihezi University, 832003 Shihezi, China. [4]Shanghai Institute of Pollution Control and Ecological Security, 200092 Shanghai, China. [5]These authors contributed equally: Yurui Fan, Haomiao Xu. ✉e-mail: xuhaomiao@sjtu.edu.cn; yyancai@sjtu.edu.cn; quzan@sjtu.edu.cn; pfxie@zju.edu.cn

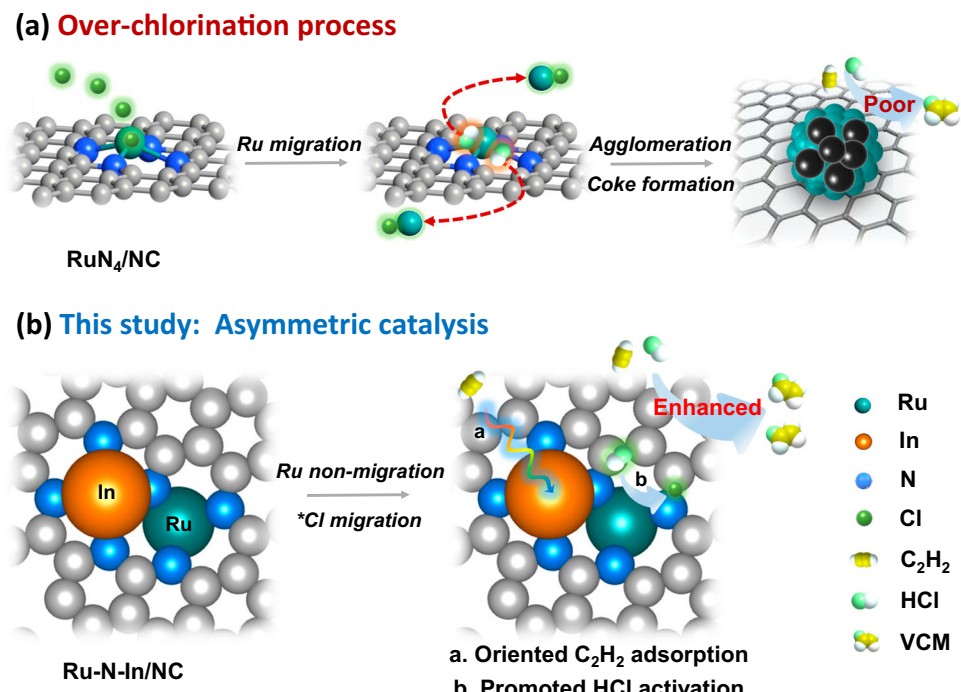

**Fig. 1 | Comparison of the RuN$_4$/NC and Ru–N–In/NC structures for acetylene hydrochlorination. a** The over-chlorination process of the symmetrical RuN$_4$/NC moiety during the hydrochlorination reaction, and **b** the corresponding solution through breaking the geometric symmetry to construct a unique asymmetric Ru–N–In/NC configuration.

empty 5*p* orbitals and exhibit a high degree of electrophilicity[31–34]. In this regard, the assembly of a single In site next to atomic Ru (i.e., In–Ru DAC, Fig. 1b) is expected to manipulate the electron configuration of Ru and deliver a desired environment to balance the trade-off between reactivity and stability, resulting in highly stable acetylene hydrochlorination[35–37].

Here, we synthesized an asymmetric Ru–In dual-atom catalyst (denoted as Ru–N–In/NC), in which Ru and In atoms are confined by the nitrogen atoms doped in carbon (NC) through metal-N bonding coordination. Accordingly, the derived Ru–N–In/NC catalyst has achieved >99.5% activity and >600 h stability for the acetylene hydrochlorination. Experiments and density functional theory (DFT) calculations jointly reveal the continuous generation of high-valence Ru species and independent adsorption sites for C$_2$H$_2$ and HCl that are responsible for the enhanced performances compared to RuN$_4$/NC (Ru is located at the center of symmetrical planar with four N-coordinated structures). Indium atoms promote stable catalysis by decreasing the Cl coordination number to prevent over-chlorination and coke deposition on atomic Ru sites. Eventually, acetylene hydrochlorination proceeds favorably over the Ru–N–In/NC structure via the Langmuir–Hinshelwood mechanism with an unconventional rate-determining step (RDS) of *CH$_2$=CH formation, as compared to the addition of *Cl on the RuN$_4$/NC.

## Results

### Synthesis and characterization of the asymmetric Ru–N–In single-atom pairs

The Ru–N–In/NC catalyst was synthesized via an atomic interface regulation strategy as illustrated in Supplementary Fig. 1[38]. Typically, nitrogen-doped carbon, ruthenium acetylacetonate (Ru(acac)$_3$), and MIL-68(In) were ground thoroughly to give a uniform mixture, which was then subjected to calcination under an Ar atmosphere. The single-metal counterparts of RuN$_4$/NC and InN$_4$/NC were prepared using the same procedures but without MIL-68(In) or Ru(acac)$_3$, respectively. Herein, the nitrogen-doped carbon supports for anchoring metals

were obtained through polyaniline (PAN) pyrolysis at 800 °C (Supplementary Figs. 2–4). Inductively coupled plasma optical emission spectrometry (ICP-OES) determined that the contents of Ru and In were ~0.98 and 0.19 wt%, respectively, which were close to the nominal loadings (Supplementary Table 1).

The transmission electron microscopy (TEM) image presents that the Ru–N–In/NC moiety possessed a hexagonal-like shape composed of randomly arranged nanosheets (Fig. 2a), similar to the morphology of NC (Supplementary Fig. 5), RuN$_4$/NC (Supplementary Fig. 6), and InN$_4$/NC (Supplementary Fig. 7). The selected area electron diffraction (SAED) pattern suggests the absence of metallic nanoparticles and low crystallinity of the carbon matrix with abundant carbon defects[39]. The C/N ratio derived from CHNS analysis was 6.52 for Ru–N–In/NC, lower than that of NC (6.73), demonstrating that Ru–N–In/NC had more carbon vacancies and unsaturated nitrogen sites (Supplementary Fig. 10)[40]. Energy dispersive spectroscopy (EDS) mappings confirmed that Ru, In, and N were uniformly distributed on the carbon matrix (Fig. 2b). Powder X-ray diffraction (XRD) patterns show all the samples had two diffraction peaks of (002) and (100) facets (Fig. 2c)[41]. The aberration-corrected high-angle annular dark-field scanning TEM (AC-HAADF-STEM) image exhibits that Ru and In atoms were well-dispersed as atomic pairs throughout the whole matrix, in which dimeric white bright spots (marked by yellow circles/boxes) were identified as Ru–N–In/NC moieties (Fig. 2d). The average distance of the two neighboring metal sites of Ru and In was determined as ~0.37 nm (Fig. 2e and Supplementary Figs. 6 and 7). The atomically dispersed Ru or In on RuN$_4$/NC or InN$_4$/NC was also confirmed (Supplementary Figs. 8 and 9). Subsequently, we compared the thermodynamic formation energies for six single/dimer-atom catalysts including RuN$_4$/NC, InN$_4$/NC, Ru$_2$/NC, In$_2$/NC, Ru–N–In/NC, etc., which revealed that the asymmetric Rn–N–In dual-atom structure had the lowest free energy of −2.793 eV (Supplementary Fig. 11). Additionally, N$_2$ adsorption–desorption curves indicate that Ru-N-In/NC had a high surface area of 690.13 m²/g (Supplementary Fig. 12 and Supplementary Table 2). Fourier-transform infrared spectroscopy (FTIR) (Supplementary Fig. 13) and X-ray photoelectron spectroscopy

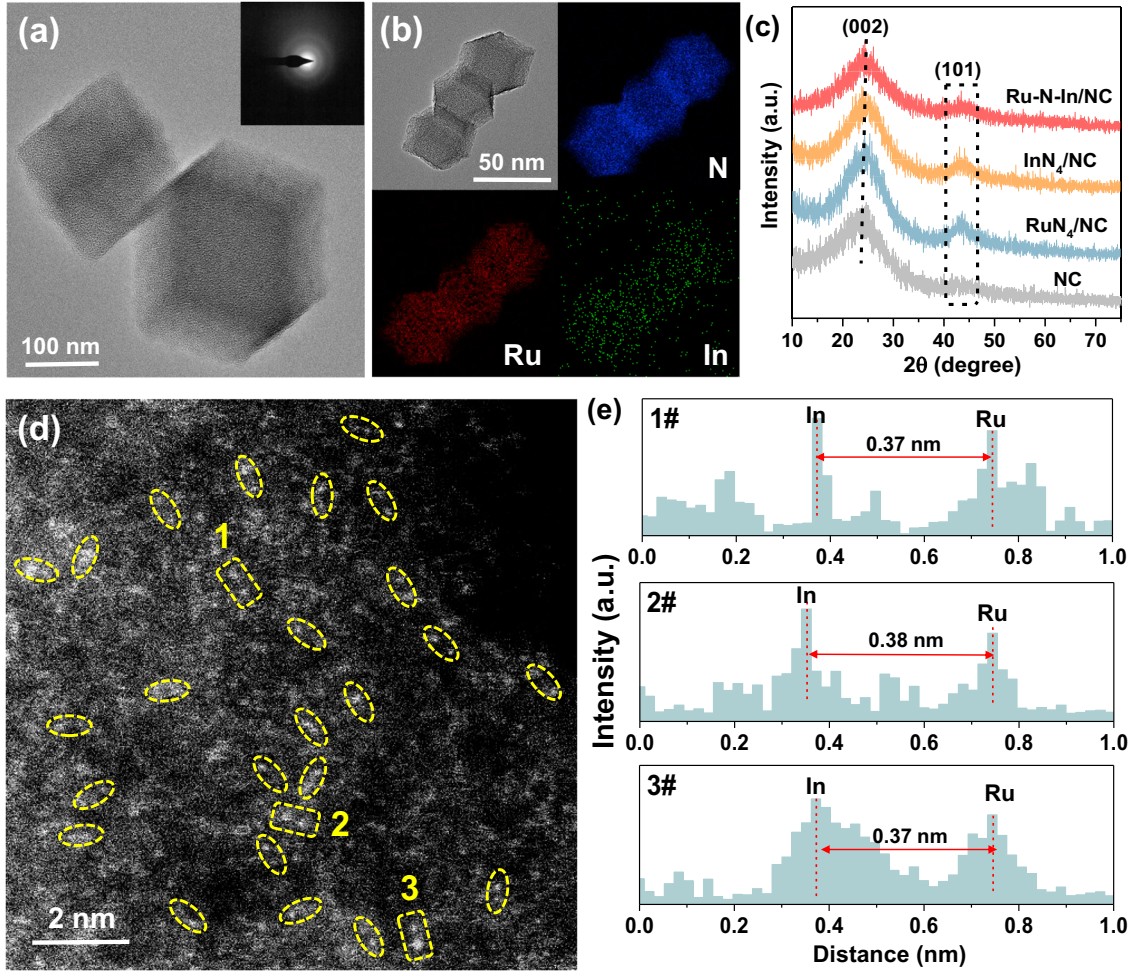

**Fig. 2 | The synthesis and structural characterization of the Ru−N−In/NC catalyst. a** and **b** TEM images of the Ru−N−In/NC moiety and the corresponding EDS mappings for Ru, In, and N, respectively. **c** Powder XRD patterns of Ru−N−In/NC, RuN₄/NC, InN₄/NC, and NC. **d** AC-HAADF-STEM image of the Ru−N−In/NC sample, in which the single-atom Ru−In pairs are highlighted by the yellow circles/boxes. **e** Atomic scale identified the distance (nm) of isolated Ru and In dual single-atom sites in the selected area with yellow rectangles 1, 2, and 3 in figure (**d**).

(XPS) (Supplementary Figs. 14–16) also further proved the formation of Ru–In coordination.

The Ru oxidation state in Ru−N−In/NC was resolved by X-ray absorption near edge spectroscopy (XANES) (Fig. 3a). The Ru K edge spectrum exhibits a pre-edge transition at 22112 eV, which falls between the peaks associated with RuCl₃ (22110 eV) and RuO₂ (22114 eV). This indicated an intermediate oxidation state between +3 and +4 for Ru in Ru−N−In/NC. The coordination structure of Ru single atom was further revealed by the extended X-ray absorption fine structure (EXAFS) analysis. Figure 3b compares Fourier-transformed EXAFS spectra for Ru−N−In/NC, RuN₄/NC, RuO₂, and Ru foil. Ru−N−In/NC exhibits first-shell scattering at 1.48 Å in R space (prior to phase correction), which is proximate to the value of 1.52 Å, found for RuO₂. This was distinct from the case for Ru foil, the first shell scattering of which is located at 2.30 Å. From these observations, we assigned the primary feature at 1.48 Å to be Ru−N bonding[42]. The atomic dispersion of Ru was also evidenced by the wavelet transform (WT) plot, with the intensity at ≈4.5 and 9.5 Å⁻¹ arising from Ru−O and Ru−Ru scattering, respectively (Fig. 3g). The quantitative structural parameter analysis based on the fitted EXAFS spectra suggested that each Ru atom coordinates with three N atoms in the first shell within the inter-plane of the tri-striazine framework, with an average bond length of 1.99 Å (Fig. 3c and Supplementary Tables 3 and 4). The atomic dispersion of Ru over the control RuN₄/NC was also investigated, where each Ru

atom was coordinated with four N atoms (Supplementary Figs. 17−20 and Supplementary Tables 3).

The coordination environments and local structures of In centers were simultaneously determined by the In K-edge XANES curves of Ru−N−In/NC, In foil, and In₂O₃ (Fig. 3d). The absorption edge of Ru−N−In/NC was centered at a higher energy than that of In₂O₃, indicating that In atoms exist at a higher valence state than in In₂O₃, which is in agreement with the result of In 3d XPS (Supplementary Fig. 14). Evidenced by the Fourier-transformed EXAFS spectra, unlike the In−In (≈2.96 Å) and In−O (≈1.67 Å) peaks observed in In foil and In₂O₃, Ru−N−In/NC exhibits a dominant peak of first-shell scattering (assigned to In−N scattering) located at ≈1.56 Å (phase-uncorrected distance) (Fig. 3e). Figure 3f presents the fitting results of the extended In K-edge, which reveals that each In atom is coordinated with four N atoms with an average In−N bond length of ≈2.15 Å (Supplementary Figs. 21−24 and Supplementary Tables 3 and 4). Moreover, the WT plots of In K-edge-weighted EXAFS indicate the atomically dispersed In species on NC supports (Fig. 3g). In addition, combining the results of DFT and X-ray absorption spectra (XAS), we further consider Ru−N−Ru/In and In−N−In/Ru as the comprehensive models and fit the raw XAS data (Fig. 3c and f). The optimal results in Supplementary Table 4 demonstrate that Ru and In are dispersed as atomic pairs on the NC supports, with a higher possibility of the presence of Ru−N−In heterostructure due to the lowest free

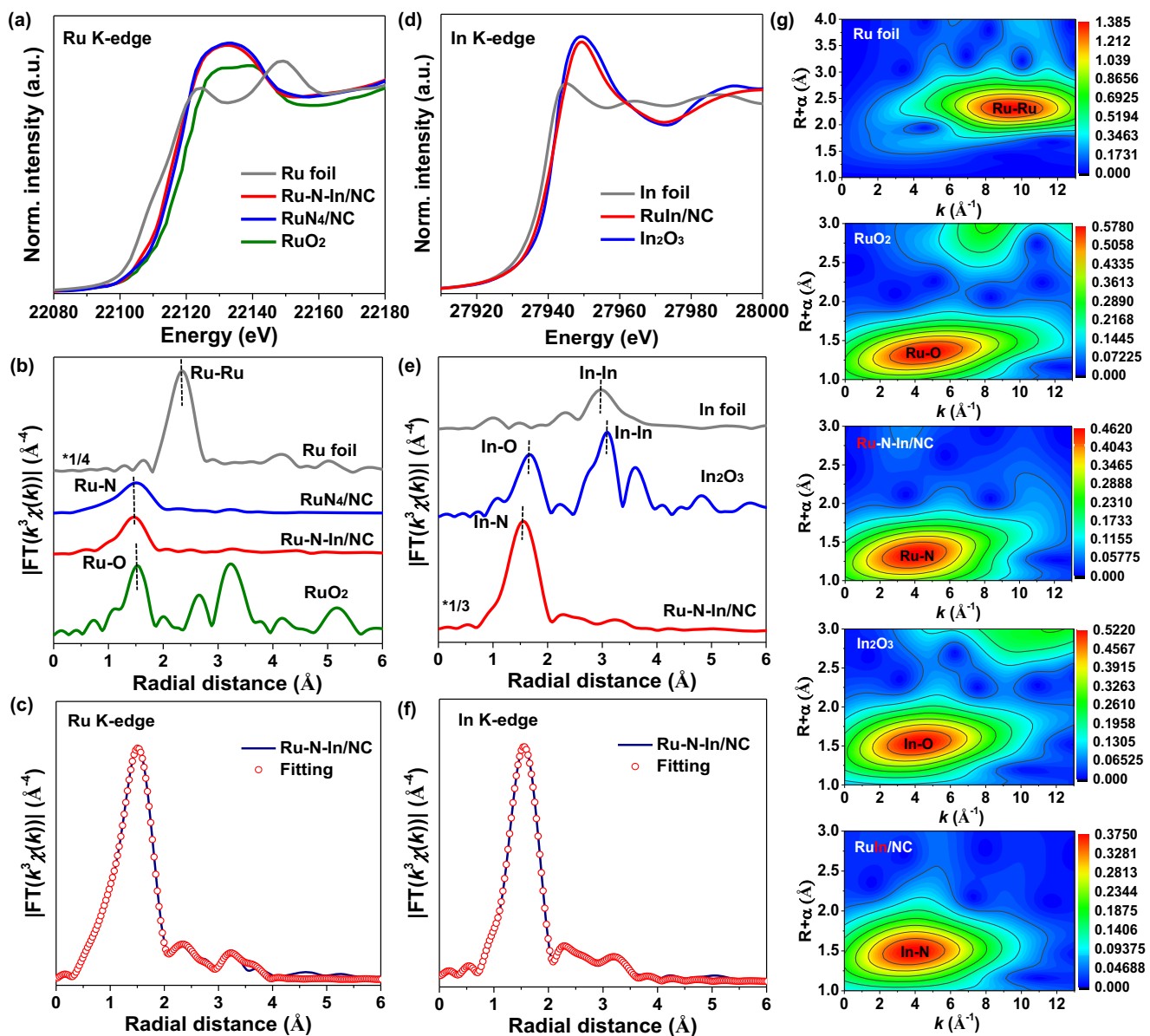

**Fig. 3 | Chemical state and atomic structure of Ru and In in Ru−N−In/NC.** Ru K-edge XAFS analysis of Ru−N−In/NC and RuN₄/NC: **a** Normalized XANES, **b** Fourier-transform XAFS spectra, and **c** XAFS curves fitting in R space. In K-edge XAFS analysis of Ru−N−In/NC: **d** Normalized XANES, **e** Fourier transform XAFS spectra, and **f** EXAFS curves fitting in R space. **g** Wavelet transform contour plots of Ru−N−In/NC at Ru K-edge and In K-edge.

formation energy of −2.793 eV compared to other counterparts (Supplementary Fig. 11).

The partial density of states (PDOS) results were compared for Ru−N−In/NC and RuN₄/NC to obtain the orbital hybridization information induced by the electronic structures of asymmetric coordination. As illustrated in Supplementary Fig. 25, the energy of the In−(p) band matches well with that of the Ru−(d) band relative to the Fermi level, indicating that a strong d−p orbital hybridization interaction occurred between Ru and In atoms. Such a d−p hybridization interaction leads to lower orbital energy caused by enhanced electron delocalization and electron redistribution. In addition, charge density analysis provided more evidence for the Ru centers in Ru−N−In/NC moiety that exhibited significant electronic delocalization accompanied by the asymmetric electronic redistribution (Supplementary Fig. 26). Compared with RuN₄/NC (−0.92 eV), the reduced Bader charge of the Ru in Ru−N−In/NC (−0.98 eV) indicated that partial electrons from the Ru centers were extracted after In introduction. The reduced electron density is also proved by the up-shifted binding

energy (-0.3 eV) of Ru 3p in Ru−N−In/NC compared to that in RuN₄/NC according to the XPS characterization (Supplementary Fig. 15).

From the above discussion, the distinct atomic structures of N-bridged Ru, In dual-atom Ru−N−In/NC catalyst have been clearly unveiled. We hypothesize that the electronic chemical configuration derived by d(Ru)−p(In) hybridization as compared to RuN₄/NC would lend them exquisite catalytic performance for acetylene hydrochlorination.

## Enhanced catalytic performance for acetylene hydrochlorination

The catalytic performances of the Ru−N−In/NC catalysts were evaluated in a continuous flow fixed-bed microreactor (Supplementary Fig. 27). The optimal reaction temperature, Ru/In ratio, gas hourly space velocity (GHSV), and carrier N-content were determined as 180 °C, 5, 180 h⁻¹, and 5.86 wt%, respectively (Supplementary Figs. 28–32). Compared with the lower acetylene conversions of RuN₄/NC (-76.35%), InN₄/NC (-18.15%), and NC (-15.69%), Ru−N−In/NC

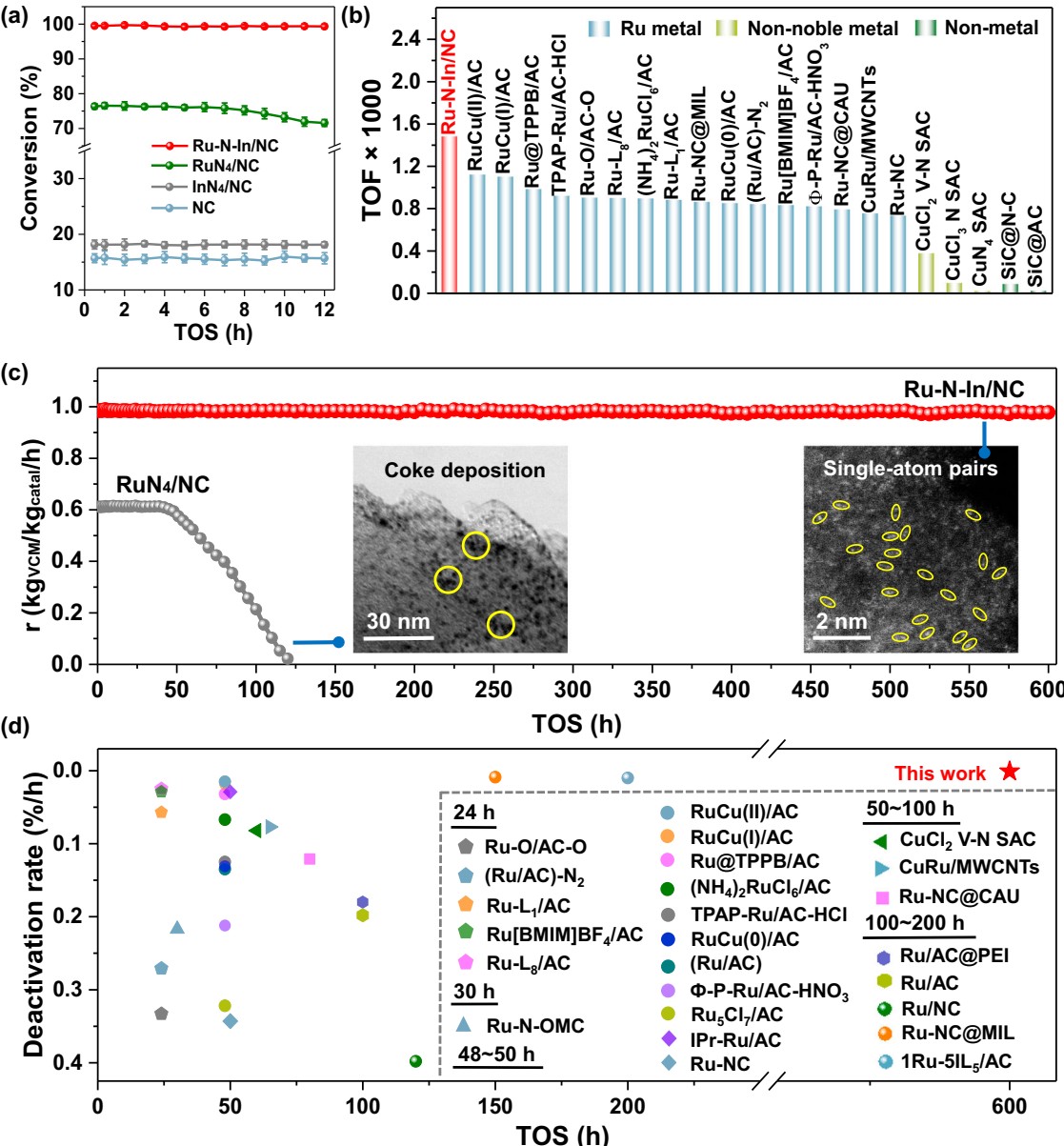

**Fig. 4 | The catalytic performance for acetylene hydrochlorination. a** $C_2H_2$ conversions of Ru−N−In/NC, RuN$_4$/NC, InN$_4$/NC and NC [Reaction conditions: $T = 180\,°C$, $P =$ ambient pressure, GHSV($C_2H_2$) = 180 h$^{-1}$, $V$(HCl)/$V$($C_2H_2$) = 1.15]. The error bars indicate the standard deviations of three experimental measurements. **b** Comparison of TOF (mol$_{C2H2}$/mol$_{metal}$/h) of Ru−N−In/NC with other Ru-based, non-noble metal and non-metal catalysts[6,7,59–67]. Note that all the obtained data are the same as our reaction conditions. The data are calculated at 180 °C, ~5% $C_2H_2$ conversion to eliminate the influence of internal and external diffusion. Each point was determined by an isolated test to eliminate the interference of catalyst deactivation. **c** Long-term catalytic performances of Ru−N−In/NC and RuN$_4$/NC, and corresponding AC HAADF-STEM images for the used samples. [Reaction conditions: $T = 180\,°C$, $P =$ ambient pressure, $V_{cat} = 1.2$ mL, $V$(HCl)/$V$($C_2H_2$) = 1.15, GHSV($C_2H_2$) = 180 h$^{-1}$.] **d** Comparison of long-term stability (TOS and deactivation rate) for Ru−N−In/NC with other recently reported catalysts.

showed an excellent initial activity of ~99.51% with negligible deactivation (Fig. 4a). Based on Fig. 4a, it can be concluded that Ru was the main active center. The introduction of In atoms apparently played a crucial role in stabilizing Ru, as evidenced by the deactivation rate of RuN$_4$/NC (~4.78%) compared to no discernible drop of acetylene conversion found for Ru−N−In/NC. Notably, the VCM selectivity for these catalysts was more than 99% except for InN$_4$/NC (Supplementary Fig. 33). The main by-product, which may originate from the oxidation of HCl or further hydrochlorination of vinyl chloride, was identified as dichloroethane (1,1-dichloroethane and 1,2-dichloroethane) through a gas chromatograph (Supplementary Fig. 34)[6,43]. To investigate the intrinsic reactivity of the aforementioned catalysts, the kinetics study was evaluated by considering the Weisz–Prater criterion and the Mears criterion to avoid the

interference of diffusion (Supplementary Texts 1 and 2)[44]. The lower apparent activation energy ($E_a$) (46.89 kJ/mol) (Supplementary Fig. 35) and Ru loss ratio (0.2%) (Supplementary Table 1) of Ru−N−In/NC compared to 65.02 and 79.31 kJ/mol for RuN$_4$/NC and InN$_4$/NC contribute to its higher catalytic activity. Further, the reaction orders of HCl and $C_2H_2$ were derived from kinetic studies, and a higher reaction order for HCl (0.67–0.92) was obtained in comparison with $C_2H_2$ (0.65–0.83) (Supplementary Fig. 36).

Subsequently, we compared the turnover frequency (TOF, mol$_{C2H2}$/mol$_{metal}$/h) of Ru−N−In/NC with the recently reported catalysts (Supplementary Table 5). Intriguingly, the activity of Ru−N−In/NC was not only far exceeded by those of other Ru-based catalysts (Fig. 4b), but also higher than that of several Au- and Pt-based samples (Supplementary Fig. 37). Moreover, the successful synthesis and

application strategy of Ru–N–In/NC is generally applicable to other noble metals (i.e., AuIn/NC, PdIn/NC, and PtIn/NC), which exhibit promising reactivities for acetylene hydrochlorination. Herein, the Ru–N–In/NC still has better activity compared to other catalysts (Supplementary Fig. 38). In addition, the activity of Ru–N–In/NC also outperforms that of RuIn/AC (activated carbon (AC) is the common support for commercial catalysts) (Supplementary Fig. 39) and commercial $HgCl_2$/AC moieties (Supplementary Fig. 40).

The long-term stability of Ru–N–In/NC and $RuN_4$/NC was subsequently evaluated at the GHSV of $180 \, h^{-1}$, where Ru–N–In/NC can maintain stable catalysis with an admirable VCM productivity of ~$0.98 \, kg_{vcm}/kg_{catal.}/h$ for more than 600 h (Fig. 4c). Such stable catalysis for Ru–N–In/NC in terms of time on stream (TOS, 600 h) and deactivation rate (0.001%/h) in the present study was apparently superior to other recently reported catalysis systems (Fig. 4d). However, the completed deactivation was observed for $RuN_4$/NC within 120 h (Fig. 4c and Supplementary Fig. 41). From the perspective of practical application, the concomitant $CO_2$ (~250 ppm) was pulsed to evaluate the tolerance of Ru–N–In/NC, in which the VCM productivity was attenuated to ~$0.76 \, kg_{vcm}/kg_{catal}/h$ due to the competitive adsorption between $CO_2$ and the reaction gases, however, the stability was still kept at 600 h (Supplementary Fig. 42)[45]. We also synthesized and tested the stability of RuIn/AC, which resulted in decreased activity (~$0.81 \, kg_{vcm}/kg_{catal}/h$) and poor stability (450 h) (Supplementary Fig. 43). The results of XRD (Supplementary Fig. 44) and XPS (Supplementary Fig. 45) indicated that the structure of post-hydrochlorination Ru-N-In/NC was perfectly intact. TEM (Supplementary Fig. 46) and AC-HAADF-STEM (Fig. 4c) images show that the used sample retained the original morphology of randomly stacked nanosheets and, most importantly, the atomically dispersed Ru–In pairs.

## Mechanisms of acetylene hydrochlorination on asymmetric Ru–N–In/NC

Prior to the mechanism investigations, the adsorption of $C_2H_2$ or HCl on asymmetric Ru–N–In/NC was studied. Temperature-programmed-desorption (TPD) experiments, in which all the desorption peaks are located at ~205 °C (Fig. 5a), were carried out to determine the adsorption sites for acetylene molecules. Compared with $RuN_4$/NC (2329.7), the higher desorption peak area for $InN_4$/NC (6653.7) proved that In atoms were more favorable for capturing $C_2H_2$ (Supplementary Table 6). As shown in Fig. 5b and Supplementary Fig. 47, In-situ diffuse reflectance infrared Fourier transform spectroscopy (DRIFTS) for $C_2H_2$ adsorption also offers a similar conclusion, in which the $\nu_{as}(C_2H_2)$ of $3258 \, cm^{-1}$ band underwent a downward shift with reference to the gas-phase value at $3287 \, cm^{-1}$, indicating that the $C_2H_2$ molecule was subjected to strong chemisorption of In–CHCH due to its significant bond polarization[32,46]. The –C≡C– bond cleavage into –C=C– bond of $C_2H_2$ upon adsorption on In species, forms $p$-π-bonded HC=CH species[47]. The integrated crystal orbital Hamilton population (ICOHP) was carried out to analyze the adsorption of $C_2H_2$ on Ru–N–In/NC and $RuN_4$/NC (Supplementary Fig. 48)[48,49]. The up ICOHP in Ru–N–In/NC decreases from −1.58 to −2.24 compared with $RuN_4$/NC, and the down ICOHP decreases from −1.76 to −2.26, demonstrating the stronger adsorption strength of the $C_2H_2$ molecules on Ru–N–In/NC. Furthermore, the $C_2H_2$ adsorption energies ($E_{ads} (C_2H_2)$) on Ru–N–In/NC, $RuN_4$/NC, and $InN_4$/NC were calculated as −0.94, −0.78 and −1.18 eV, which aligned with the trend found in TPD (Supplementary Figs. 49 and 50). The modest $C_2H_2$ adsorption strength allowed the optimal interaction of $C_2H_2$ with Ru–N–In/NC ($R^2 = 0.87$), which is required to trigger the reaction (Supplementary Fig. 51) compared to the single atom counterparts.

The HCl-TPD curves show that the desorption temperature of the three samples is located at ~170 °C. A remarkable decline of the desorption peak area for Ru–N–In/NC (4467.3) was observed compared to

$RuN_4$/NC (5474.3), but still higher than that of $InN_4$/NC (2638.7), indicating that the addition of In atoms weakened the adsorption of HCl on the Ru site in the case of Ru–N–In/NC (Fig. 5c and Supplementary Table 6). Further, we compared the adsorption energy of Ru and In over Ru–N–In/NC. Lower adsorption energy (−0.63 eV) over Ru sites was obtained, in which HCl was adsorbed via the interaction of Ru⋯Cl–H between Cl (Lewis base) and Ru (Lewis acid) (Fig. 5d and Supplementary Fig. 52). In contrast, stronger HCl adsorption over $RuN_4$/NC ($E_{ads}$ (HCl) = −0.83 eV) was observed, demonstrating a significant over-chlorination potential for Ru centers (Supplementary Fig. 53). In summary, the dimeric Ru-In geometry with proper distance (~0.37 nm) enables the independent adsorption configuration for $C_2H_2$ and HCl and mediates the steric hindrance for that, accounting for the enhanced activities compared to $RuN_4$/NC.

Subsequently, we investigated the structure-activity relationship for the Ru–N–In/NC dual-atom catalyst. Generally, the catalyst with higher adsorption energy for the reactants has higher catalytic activity. However, as depicted in Fig. 5e, Ru–N–In/NC owns the medium adsorption strengths for reactants, giving rise to the highest level of reactivity[50]. Next, an attempt was undertaken to correlate the $d$-band center with TOF, which exhibited a remarkable linear relationship ($R^2 = 0.91$) (Fig. 5f and Supplementary Fig. 54). Hence, it could be understood that the strong $d$(Ru)−$p$(In) interaction results in the up-shift of $d$-band center of Ru to Fermi level, forming active sites with unique electron configurations for the appropriate adsorption strengths of reactants on the atomic Ru and In sites that improve the intrinsic activity of Ru–N–In/NC[51].

Figure 5g schemes the Gibbs free energy diagrams of acetylene hydrochlorination over the $RuN_4$/NC and Ru–N–In/NC catalysts based on the first principle, and this process follows the Langmuir-Hinshelwood co-adsorption mechanism. Differently, $C_2H_2$ and HCl are simultaneously adsorbed on the Ru atom of $RuN_4$/NC, whereas $C_2H_2$ and HCl molecules are separately adsorbed on the In and Ru atoms of Ru–N–In/NC, eventually forming the co-adsorbed $(C_2H_2 + HCl)_{ads}$ state. Subsequently, the HCl molecule begins to migrate, and the H–Cl bond is continuously elongated and broken, causing the *H atom to be endothermically added to *$C_2H_2$ to form *CH=CH. Due to the diminished steric hindrance for the Ru–N–In/NC in comparison with $RuN_4$/NC, this process (energy barrier = 0.54 eV) serves as the new rate-determined step (RDS) for Ru–N–In/NC moiety. In the following step, the Cl* atom approaches another C atom of *$C_2H_2$, and then coordinates to *CH=CH to generate *$CH_2CHCl$ over Ru centers. For the $RuN_4$/NC moiety, this endothermal process can be regarded as the RDS, while the Ru–N–In/NC decreases the energy barrier of *Cl addition from 0.61 to 0.21 eV. Overall, the cooperation of In atoms changes the RDS of acetylene hydrochlorination by regulating the steric hindrance effect of atomic active sites, skipping the conventional barrier limitation of *Cl addition for the $RuN_4$/NC surfaces. Eventually, the reaction proceeds at a lower free energy on Ru-N-In/NC with the pathway of $(C_2H_2 + HCl)_{ads} \rightarrow {}^*C_2H_2 + {}^*HCl \rightarrow {}^*CH= CH \rightarrow {}^*CH_2CHCl \rightarrow CH_2CHCl$.

## Intrinsic understanding of the high stability of asymmetric Ru–N–In/NC

To unlock the intrinsic mechanism of the promising stability of Ru–N–In/NC, we carefully investigated the chlorination processes of Ru–N–In/NC and $RuN_4$/NC. At the beginning, the dissociation energy of Ru was calculated as 1.86 and 1.27 eV for Ru–N–In/NC and $RuN_4$/NC, respectively, suggesting a higher binding strength between single atom Ru and the substrate in the case of a dual-atom catalyst (Fig. 6a). Then, in-situ DRIFTS $NH_3$ adsorption characterization was carried out to further determine Lewis acidity. The peak of $1591 \, cm^{-1}$ attributed to the $NH_3$ adsorption on Lewis acid sites, in which the intensity of this peak for Ru–N–In/NC was remarkably faded, demonstrating that the Lewis acidity of Ru–N–In/NC was weakened compared to $RuN_4$/NC

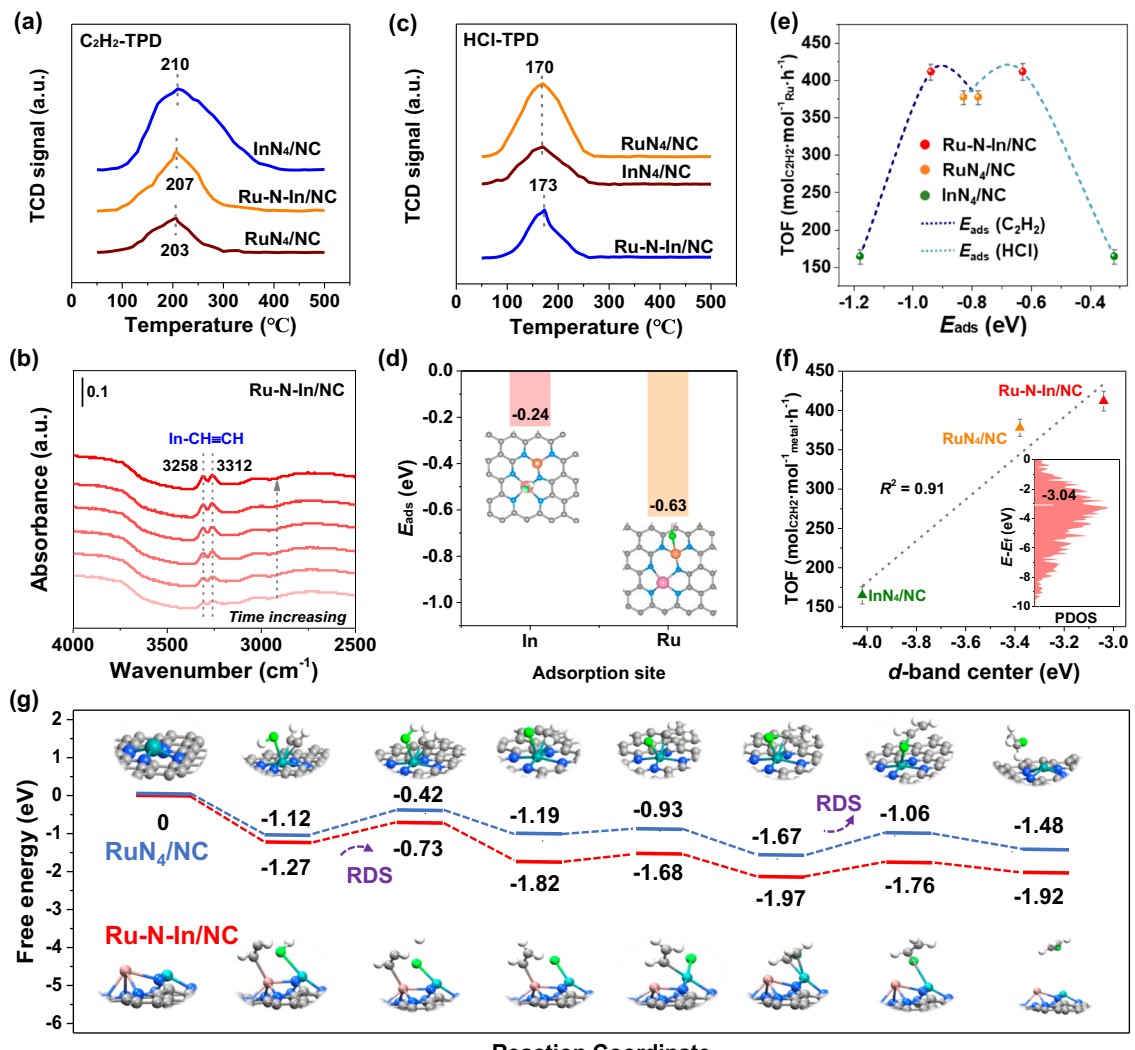

**Fig. 5 | The study of the catalytic mechanism for acetylene hydrochlorination.**
**a** $C_2H_2$-TPD curves of the Ru−N−In/NC, RuN$_4$/NC, and InN$_4$/NC samples. **b** In-situ
DRIFTS of $C_2H_2$ adsorption profiles over the Ru−N−In/NC surfaces. **c** HCl-TPD
curves of the Ru−N−In/NC, RuN$_4$/NC, and InN$_4$/NC. **d** Comparison of adsorption
energy for HCl molecule on Ru and In centers. **e** The adsorption energy of $C_2H_2$/HCl
versus TOF of the Ru−N−In/NC, RuN$_4$/NC, and InN$_4$/NC. The error bars indicate the
standard deviations of three experimental measurements. **f** $d$-band center as a
descriptor versus the TOF of the Ru−N−In/NC, RuN$_4$/NC, and InN$_4$/NC, in which the
projected density of states analysis of Ru−N−In/NC was inserted. The error bars
indicate the standard deviations of three experimental measurements. **g** DFT cal-
culations of the acetylene hydrochlorination reaction pathways over the Ru−N−In/
NC and RuN$_4$/NC moieties. Cyan, orange, green, blue, gray, and white spheres
represent Ru, In, Cl, N, C, and H atoms, respectively.

(Supplementary Fig. 55), which could potentially prevent over-
chlorination on the atomic Ru site. Further, scanning electron micro-
scopy (SEM) and EDS images of the used Ru−N−In/NC and RuN$_4$/NC
samples indicate that the enrichment of Cl* on the catalyst surface is
noticeably reduced from 14.41 to 4.08 wt% (Supplementary Fig. 56).

Next, we utilized Ru K-edge XANES to track the chemical and
geometric states of Ru after chlorination for three hours for
Ru−N−In/NC and RuN$_4$/NC. The pre-edge transition of Ru−N−In/NC-
3h is between RuCl$_3$ and RuO$_2$ references, suggesting that the Ru
valence states were almost the same (between +3 and +4) before and
after the reaction. In contrast, the white-line intensity of chlorinated
RuN$_4$/NC decreased with the increasing exposure time, indicating
that the Ru chemical state gradually reduced (Fig. 6b). The FT $k^3$-
weighted $\chi(k)$-function and structural parameters analysis based on
the fitting of EXAFS spectra showed that the coordination number of
Ru−Cl over Ru−N−In/NC-3h was kept ~1, while the coordination
number of Ru−Cl on the post-hydrochlorination RuN$_4$/NC presented
an increasing trend, from ~1.3 (RuN$_4$/NC-1h) to ~2.6 (RuN$_4$/NC-3h)
(Supplementary Table 7), verifying that the asymmetric Ru−N−In/NC

configuration inherently blocked the over-chlorination of Ru. AC-
HAADF-STEM analysis of RuN$_4$/NC-3h indicated that the Ru centers
aggregated from single atoms to nanoparticles with sizes up to
~4 nm, whereas the single-atom pairs were still retained on Ru−N−In/
NC-3h catalyst (Supplementary Fig. 57), and even on Ru−N−In/NC-
600h catalyst (Fig. 4c). Further, the XPS spectra of Cl 2$p$ for the post-
hydrochlorination samples show an apparent increase in the surface
Cl content than that of the fresh sample for RuN$_4$/NC-3h (Supple-
mentary Fig. 58 and Supplementary Table 8).

To obtain insights into bonding information for Ru-Cl interac-
tions, we conducted PDOS analysis for Ru $d$ orbitals before and after
Cl* adsorption. Generally, the H$^{\delta+}$ and Cl$^{\delta-}$ species in HCl can be seen as
Brønsted acid and Lewis base, respectively. Thus, the Ru atom (Lewis
acid) can accept the Cl$^{\delta-}$ atom (Lewis base) of HCl to form Ru−Cl$^{\delta-}$ and
promote the scission of the H−Cl bond[52]. As illustrated in Fig. 6c and
Supplementary Fig. 59, upon the adsorption of Cl*, the interaction
between the $p$ orbital of Cl and the $d_{z^2}$ orbital of Ru gives the σ bonds on
both Ru−N−In/NC and RuN$_4$/NC. However, the π bonds are derived
from the coupling between the $p$ orbital of Cl and the $d_{x^2-y^2}$ orbital of

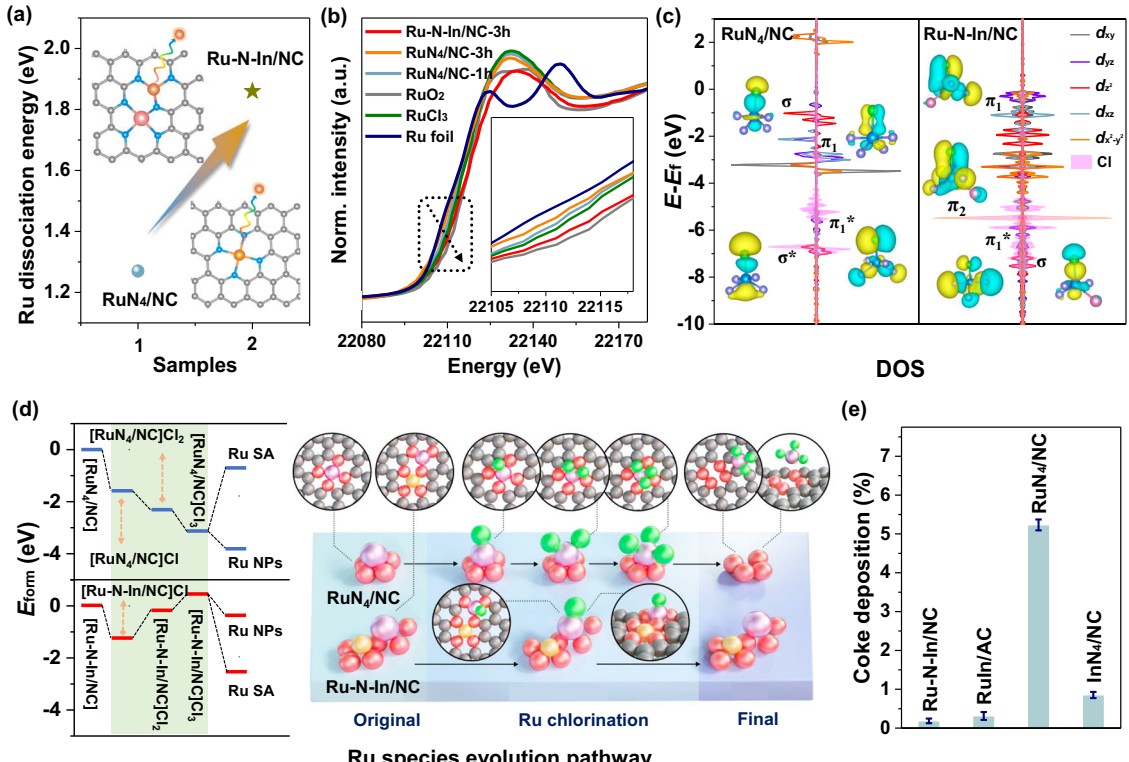

**Fig. 6 | Stabilization of the Ru center by decreasing its chlorination degree. a** Ru dissociation energy of Ru–N–In/NC and RuN$_4$/NC moieties. Gray, blue, green, orange, and pink balls represent C, N, Cl, Ru, and In atoms, respectively. **b** The normalized Ru K-edge XAFS plots of the post-hydrochlorination samples of Ru–N–In/NC-3h, RuN$_4$/NC-3h, and RuN$_4$/NC-1h, in which the Ru foil, RuCl$_3$, and RuO$_2$ spectra serve as references. **c** Projected density of states of Ru $p$ orbitals and Cl* $d$ orbitals after Cl* adsorption over the interface of RuN$_4$/NC and Ru–N–In/NC. σ and σ* represent the bonding and antibonding between $d_z^2$ orbital of Ru and $p$ orbital of Cl, π$_1$ and π$_1$* represent the bonding and antibonding between $d_{yz}/d_{xz}$

orbital of Ru and $p$ orbital of Cl, π$_2$ represents the bonding between the $d_{x2-y2}$ orbital of Ru and $p$ orbital of Cl. **d** The formation energy of various Ru chlorinated species and the corresponding illustration of the evolution pathway of Ru species over the Ru–N–In/NC and RuN$_4$/NC structures. Gray, pink, purple, yellow, and green balls represent C, N, Ru, In, and Cl atoms, respectively. **e** Coke deposition of the post-hydrochlorination Ru-based catalysts, determined by the weight loss differences. The error bars indicate the standard deviations of three experimental measurements.

Ru for Ru–N–In/NC, while the involved orbitals are changed to $p$(Cl) and $d_{yz}/d_{xz}$ for RuN$_4$/NC (Fig. 6c and Supplementary Fig. 60). Such distinct hybridization manner is attributed to the asymmetric geometry and unique electron configuration of atomic Ru enabled by the strong $d$(Ru)–$p$(In) interaction for Ru–N–In/NC[53–55].

To reveal the evolution pathways of the chlorination process on the atomic Ru of Ru-N-In/NC and RuN$_4$/NC, the formation energies of chlorinated species and the related PDOS analysis were carried out. For RuN$_4$/NC, the gradual addition of *Cl to Ru centers was observed, giving a stable three Cl-coordination configuration (denoted as [RuN$_4$/NC]Cl$_3$) eventually. Most importantly, the entire chlorination process from RuN$_4$/NC to [RuN$_4$/NC]Cl$_3$ is highly exothermic (Fig. 6d). That is because once the original symmetrical confinement of RuN$_4$/NC was broken with the adsorption of the first Cl*, the subsequent chlorination process became more energy favorable. In contrast, although the evolution of [Ru–N–In/NC]Cl was easily obtained, the addition of second Cl* turned to be highly endothermic, as evidenced by the emergence of several obvious antibondings, such as the σ* bond ($p$ orbital of Cl and $d_z^2$ orbital of Ru) and π* bond ($p$ orbital of Cl and $d_{x2-y2}$ orbital of Ru) shown in the PDOS plots of the [Ru–N–In/NC]Cl$_2$ atomic interface (Supplementary Fig. 61). Therefore, the over-chlorinated structures such as [Ru–N–In/NC]Cl$_2$ and [Ru–N–In/NC]Cl$_3$ ($E_{form} > 0$ eV) are unstable (Fig. 6d). Overall, the strong $d$(Ru)–$p$(In) interaction orbital couplings in Ru–N–In/NC promote the thermodynamic transition of the chlorination process from exothermal to endothermal compared to RuN$_4$/NC, intrinsically avoiding the over-chlorination of Ru and ensuring its excellent

stability. Furthermore, the progressive chlorination of Ru-N-In/NC, particularly in proximity to Ru atoms, results in a reduced electron density at the metal sites, as concluded from a shift to higher binding energies in the XPS spectra of Ru $3p$ (Supplementary Fig. 62). Unfortunately, the active Ru species of RuN$_4$/NC in comparison with Ru–N–In/NC was completely deactivated to Ru$^0$ under the reactive atmosphere (C$_2$H$_2$ + HCl), as demonstrated by Ru $3p$ XPS (Supplementary Figs. 63–65).

Coke accumulation is an important indicator for measuring the stability of catalysts, thus, thermogravimetry (TG) was applied to determine coke deposition through the weight loss differences[56,57]. Fortunately, slight coke deposits (ca. 0.18%) were observed on the surface of Ru–N–In/NC, while the ample coke coverage (ca. 5.23%) was accumulated over the RuN$_4$/NC surface (Fig. 6e). Thus, the possible explanation for the observed changes from "coking-prone" carbon to "coking-resistant" carbon upon indium introduction is that Ru–N–In/NC not only effectively adsorbed acetylene but also was favorable for the desorption of VCM[13]. Furthermore, Ru–N–In/NC can restrain excessive Cl* deposition and addition on the catalyst surface, thus alleviating the aggregation of Ru active sites and preventing the coupling of C–C bonds to form coke precursors. However, for RuN$_4$/NC, excessive Cl* adsorption induced the aggregation of Ru, leading to the shift in the predominating deactivation mode, from agglomeration to coke deposition[15]. In addition, the leaching experiment and extraction operation interpreted that the Ru–N–In/NC catalyst also had superior recyclability and reusability (Supplementary Figs. 66 and 67). In conclusion, the successful design of Ru–N–In/NC catalyst with multiple

excellent properties opens up the possibility of practical application of single-atom Ru catalysts.

## Discussion

In summary, we developed an asymmetric Ru–N–In/NC dual single-atom catalyst by breaking symmetrical RuN₄/NC geometry to efficiently address the over-chlorination issue of Ru for highly active and stable acetylene hydrochlorination. The experiments and theoretical analysis jointly revealed that the enhanced catalytic performances were caused by the optimized electron arrangement and local density-of-states distribution of the Ru·In synergistic centers. Accordingly, Ru–N–In/NC exhibited exceptional $C_2H_2$ conversions (>99.5%) and long-term stability (>600 h), which was evidently far superior to that of the symmetrical RuN₄/NC counterpart, state-of-the-art metal-based moieties, and commercial $HgCl_2$ catalysts. The enhanced performances result from the electronic delocalization and electron transfer induced by $d$-$p$ hybridization between Ru and In atoms. The orbital couplings promote the thermodynamic transition of Cl* intermediates interacting with Ru from endothermal to exothermal, lowering the energy barrier for Cl* addition. Eventually, the selective adsorption of $C_2H_2$ and HCl on different sites achieves the transformation from "coking-prone" carbon to "coking-resistant" carbon, suppressing coke accumulation. The presented findings provide insight into the rational design of Ru SACs with excellent acetylene hydrochlorination performance.

## Methods

### Catalyst preparation

**Preparation of NC.** The N-doped carbon supports were prepared through a two-step synthesis, including the oxidative polymerization of aniline and a subsequent carbonization step[9]. 50 mmol of aniline was dissolved in 40 mL of deionized water with a pH of 0.4 and cooled to 4 °C. Then, the mixture was added to the pre-cooled $(NH_4)_2S_2O_8$ solution (5 mol/L, 4 °C) under vigorous stirring. To form higher molecular weight PAN, the polymer slurry was continuously stirred for 24 h at room temperature to complete the polymerization process, subsequently filtered, and washed with deionized water and ethanol. The formed PAN was dried in static air at 120 °C for 12 h and finally carbonized at 800 °C (5 °C/min) in an $N_2$ atmosphere for 1 h to yield NC.

**Preparation of MIL-68(In).** MIL-68(In) was employed as the precursor of In source[58]. Briefly, 0.450 g of 1,4-dicarboxybenzene and 0.350 g of $In(NO_3)_3\cdot 5H_2O$ were dissolved in 135 mL of N,N-Dimethylformamide (DMF). The mixture was ultrasound for 10 min and then heated at 120 °C for 40 min under constant stirring. The resulting solid product was collected through centrifugation and washed in a mixture of DMF and $CH_3OH$ (30 mL, $v/v$ = 3:1) five times, before being dried at 60 °C for 12 h under vacuum to get white MIL-68(In).

**Preparation of Ru–N–In/NC.** The catalysts were prepared through an incipient wetness impregnation method (Ru loading of 1.0 wt%). Briefly, 5.0 g of NC and 0.1026 g of $Ru(acac)_3$ were added with 50 mL of deionized water and 25 mL of ethanol. The mixture was sonicated for 30 min, and heated at 90 °C under stirring to evaporate the water. The obtained mixture was dried under vacuum at 60 °C for 12 h. The dried powder was then mixed with a certain amount (adjust the adding amount according to the load of In element) MIL-68(In) and ground for 30 min to form a homogeneous mixture. Then, the mixture was calcinated at 900 °C for 3 h at a heating rate of 5 °C/min under an Ar atmosphere. The obtained sample was denoted as Ru–N–In/NC. The synthesis of RuN₄/NC and InN₄/NC followed a similar procedure as that of Ru–N–In/NC, except that MIL-68(In) and $Ru(acac)_3$ were not used for the synthesis of RuN₄/NC and InN₄/NC, respectively.

**Preparation of RuIn/AC.** The first step of preparing the RuIn/AC catalyst is to clean the commercial activated carbon to obtain pure AC supports[7]. Briefly, the commercial activated carbon (CAC) was cleaned with 36% HCl to remove ash and impurity ions on the surface, followed by washing with ultrapure water to neutral pH and desiccating at 150 °C for 24 h. Then, the washed CAC was immersed into the ammonium persulfate solution (5 mol/L) in 1.0 M of $H_2SO_4$ at room temperature for 24 h. After oxidation treatment, CAC was washed with distilled water until pH neutral then maintained at 60 °C in the thermostat water bath for 12 h and desiccated at 120 °C for 12 h to obtain AC. Then, 1.0 g of AC was added to the mixed solution (10 mL of deionized water and 5 mL of ethanol) containing 20.6 mg of $Ru(acac)_3$ and 5.6 mg of $In(NO_3)_3\cdot 5H_2O$. The mixture was sonicated for 30 min, and heated at 90 °C under stirring to evaporate the water. The resulting mixture was calcinated at 900 °C for 3 h at a heating rate of 5 °C/min under an Ar atmosphere to obtain the RuIn/AC catalyst.

### Catalyst characterization

Powder XRD patterns were acquired using a powder X-ray diffractometer (PANalytical, Netherlands) with Cu-Kα radiation ($\lambda$ = 1.54060 Å). Laser Raman spectra were obtained on a LabRAM HR800 Evolutions (Horiba, Japan) spectrometer. Fourier transform infrared (FTIR) spectroscopy was recorded using a Nicolet 6700 FT-IR spectrometer, and the spectra were recorded at every certain time by accumulating 64 scans with a resolution of 4 cm⁻¹. The chemical composition and relative contents of the catalyst samples were acquired by XPS equipped with Al-Kα radiation (Thermo Fischer, ESCALAB 250Xi). ICP-OES was carried out on an Agilent ICP-OES 5110 instrument. Brunaur–Emmett–Teller surface areas were acquired with a surface area analyzer (Quantachrome NOVA 2200E). TG analysis was carried out from 50 to 800 °C by using a Mettler Toledo in an air atmosphere with a heating rate of 10 °C/min. TPD experiments were measured from 30 to 600 °C with a 10 °C/min heating rate by Quantachrome Instruments (AMI-90) in a He atmosphere. The $H_2$-TPR experiment was performed on an automatic adsorption instrument (AutoChem II 2920). The catalyst morphology was characterized by TEM, and EDS images were also obtained. HR-TEM and AC-HAADF-STEM were performed on a double-corrected microscope JEM-ARM200F (GrandARM, JEOL).

In-situ FT-IR spectra of acetylene adsorption were conducted on a Nicolet 6700 FT-IR spectrometer using a liquid $N_2$-cooled mercury cadmium telluride (MCT-A) detector. In-situ FT-IR spectra of $NH_3$ adsorption were recorded on a Nicolet 6700 spectrometer in the range of 600−2000 cm⁻¹ with a resolution of 4 cm⁻¹ and 64 scans. XANES and EXAFS measurements of Ru and In were collected at the beamline BL14W1 of the Shanghai Synchrotron Radiation Facility (SSRF) in a fluorescence mode at room temperature. Data reduction, data analysis, and EXAFS fitting were performed and analyzed with the Athena and Artemis programs of the Demeter data analysis packages that utilize the FEFF6 program to fit the EXAFS data. Note that the above characterization procedures are shown in detail in the Supplementary Materials.

### Catalytic performance tests

Catalytic performance evaluations were carried out in a fixed-bed microreactor (i.d. 8.0 mm) with accurately controlled temperature and gas flow. Prior to testing, a certain amount of catalyst was loaded into the reactor, and nitrogen was imported into the reactor to remove the air and water from the catalytic system. The catalyst was heated in a He flow to the desired temperature and allowed to stabilize for at least 30 min before the reaction mixture. Then, the purified HCl was introduced into the reactor to activate the catalyst at a temperature of 180 °C for 30 min with a flow rate of 10 mL/min. Subsequently, $C_2H_2$

and HCl ($V(\text{HCl})/V(\text{C}_2\text{H}_2) = 1.15$) were fed through filters with GHSV($\text{C}_2\text{H}_2$) of $180\,\text{h}^{-1}$ at $180\,°\text{C}$. The effluent gas was passed into NaOH solution and a dryer, followed by analysis with a gas chromatograph (PANNA A60, China) equipped with a flame ionization detector (FID) and a KB-624 column ($30\,\text{m} \times 0.32\,\text{mm} \times 1.8\,\mu\text{m}$)[6,9].

The catalytic performances were evaluated in terms of acetylene conversion ($C_A$), VCM selectivity ($S_{VCM}$), and turnover frequency (TOF, $\text{h}^{-1}$), which were defined and expressed as the following equations, respectively.

$$C_A = \frac{T_{A0} - T_A}{T_{A0}} \times 100\% \tag{1}$$

$$S_{VCM} = \frac{T_{VC}}{1 - T_A} \times 100\% \tag{2}$$

$$\text{TOF} = \frac{n(\text{C}_2\text{H}_2)^{\text{inlet}} - n(\text{C}_2\text{H}_2)^{\text{outlet}}}{\text{mol}_{\text{metal}} \times \text{h}} \tag{3}$$

where $T_{A0}$, $T_A$, and $T_{VC}$ represent the volume fraction of acetylene in the feed gas, the volume fraction of acetylene in the product gas, and the volume fraction of vinyl chloride in product gas, respectively. $n(\text{C}_2\text{H}_2)^{\text{inlet}}$ and $n(\text{C}_2\text{H}_2)^{\text{outlet}}$ are defined as the molar flows of $\text{C}_2\text{H}_2$ at the inlet and outlet of the reactor, respectively. The $\text{mol}_{\text{metal}}$ is the total mole number of metals on the evaluated catalysts. The h represents time on stream.

## DFT calculations

DFT calculations were conducted in this study to determine the optimum configurations, obtain the adsorption energy values, as well as verify the catalytic pathways and transition states. The calculations were performed using the Vienna Ab initio Simulation Package (VASP) with projector-augmented wave core potentials and the PBE-D3 functional. The detailed simulation processes are exhibited in the Supplementary Materials.

## Data availability

All source data supporting the findings of this study are available within the paper and the Supplementary Information files or available from the corresponding author upon request. Source data are provided with this paper.

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

## Acknowledgements

This work was financially supported by the National Key R&D Program of China (2022YFC3901100, 2022YFC3703800), National Natural Science Foundation of China (22176121, 52070129), Natural Science Foundation of Zhejiang Province (LR22B060002), and National Natural Science Foundation of China (22278365).

## Author contributions

Y.F., H.X., Y.Y., and P.X. conceived the idea and designed the research; Y.F., G.G., and L.M. prepared materials and performed characterizations; M.W. and W.H. assembled the test system; Y.F., H.X., Z.Q., and P.X. analyzed and interpreted the results; and Y.F., H.X., Y.Y., B.D., N.Y., and P.X. contributed to the writing and revising of the paper.

## Competing interests

The authors declare no competing interests.
