## [Peer Review File · Nature Communications]

Asymmetric Ru-In Atomic Pairs Promote Highly Active and Stable Acetylene HydrochlorinationREVIEWER COMMENTS

Reviewer #1 (Remarks to the Author):

To seek other metals to substitute the toxic mercuric chloride catalyst for acetylene hydrochlorination is of great importance. The Ru-based catalysts were reported to be active for this reaction and proposed to be potential candidates. In this manuscript, the authors reported a Ru single-atom catalyst (SAC) that can resist the aggregation of Ru atoms caused by over-chlorination by synthesizing the asymmetric Ru-In atomic pairs. Assisted by In, the loss of Ru could be greatly suppressed (from 4.27% to 0.20%) after reaction for 100 h. Besides, the stability of the RuIn/NC catalyst was also greatly enhanced, which could run 600 h without obvious deactivation. This result is claimed to be the most outstanding Ru-based catalyst among all of those that were ever reported in the literatures. It can be accepted after carefully considering the following questions:

1. Although the activity and stability was enhanced by synthesizing RuIn/NC SAC, the evidences for the Ru-N-In structure were not solid enough. Firstly, because the atomic number of Ru and In was close to each other (44 vs 49), the HAADF-TEM results showed two atoms with similar lightness. Besides, only three pairs of Ru/In singles atoms were tested for their distances. This result could not stand for the whole sample. The existence of the Ru-N-Ru and In-N-In could not be excluded. Secondly, the loss of In was not suppressed, which was 0.53% and 0.51% for the RuIn/NC and In/NC catalyst after reaction for 100 h, respectively. Was the loading of Ru and In tested after reaction for different time intervals (e.g., 100 h, 200 h, 300 h...) and compared to see if their loading became stable after a certain reaction time? If there were Ru-N-In interactions, would the loss of Ru and In be avoided? Thirdly, the EXAFS data could not tell the adjacent atom was In or Ru, either. In supplementary Table 4, the r-factor for the data fitting results was close to 0.02, besides, the other parameters were also reasonable. Therefore, there might be Ru-N-Ru/In (2.906 Å) and/or In-N-Ru/In (3.558 Å) interactions in the RuIn/NC sample. But it could not be the only evidence. The data fitting range of should be provided under supplementary Table 4.
2. The BET results in supplementary Table 2, there were too many significant figures. There were detection limits for the instrument, so they cannot be so precise.
3. In supplementary Table 7, why there were two r-factors for one group of data (Ru-N and Ru-Cl)?
4. In supplementary Table 8, the ratio of Ru-Cl of the Ru/NC sample decreased with reaction time, while that of the RuIn/NC sample became almost stable after reaction for 1 h. What is the possible reason?

Reviewer #2 (Remarks to the Author):

This manuscript by Fan et al. described a Ru-In dual single-atom catalyst (SAC), possessing impressive catalytic stability for acetylene hydrochlorination. Abundant characterizations and

theoretical calculations were conducted to reveal the superior catalytic performance and several interesting arguments were proposed, which might arouse interests for readers and provide insights for future SACs design. Some major concerns should be addressed though before I can recommend for the publication on Nat. Commun.

1. As is well known, the intensity of the image in STEM is proportional to the square of the atomic number. However, in Figure 1e, f and g, the intensity of Ru (44 #) atom is higher than that of In (49 #), please explain. Moreover, I am concerned whether STEM can distinguish these two elements to obtain the conclusion of dual Ru-In SAC.
2. The result shows that the optimal wt. or mol. Ru/In ratio is $\sim 5:1$ (Figure S27). If the Ru-In dual single-atom site is proposed to be the most active, why is the optimal Ru/In ratio not close to 1?
3. The characterizations in the article are based on catalysts with a Ru/In molar ratio of $\sim 5:1$. Then the authors reported the structure of dual Ru-In single atom site with a Ru-N coordination number of 3.3. What is the coordination structure of excess Ru atoms without the pair with In? According to the structure of RuN₄ in Ru/NC, the Ru-N coordination number in RuIn/NC should be closer to 4 rather than 3 if N₄ coordination is adopted (Excess Ru atoms are more than the Ru paired with In).
4. Followed by the previous one, RuIn/NC exhibited much greater stability than Ru/NC. But how to explain the stability of excess Ru without pairing with In?
5. The optimal adsorption mode of acetylene on RuIn/NC showed by DFT (Figure S46) seems to be inconsistent with the results of in-situ DRIFTS, and no chemisorption characteristics were observed by DFT. In addition, it is suggested the author consider the adsorption free energy of acetylene at the experimental temperature, which could be significantly different from the electronic energy.
6. In the calculated reaction pathway (Figure 4g), the co-adsorption configuration of C₂H₂ and HCl on Ru-In site appears unusual and inconsistent with the separately optimized configurations (Figure S46 and S49). Is this the most stable co-adsorption configuration?
7. The model in Figure 5d seems to have some issues, as N atoms do not replace C but instead appear above C atoms.
8. The author proposes a mechanism by which the chlorination of Ru single atoms leads to its migration out of the N₄ site. This is interesting. But I wonder what is the “chlorinated atmosphere” referred to by the authors in the manuscript and the caption of Figure S62, and is it HCl? If not please add an explanation in the catalytic evaluation section. Hydrogen chloride is not commonly considered as a reducing gas, why can RuN₄ be reduced to Ru by HCl, and what is the potential mechanism?
9. Note that RuIn/AC without N also has good catalytic performance and inhibition of coke deposition. Do Ru and In also exhibit single atom dispersion? How do they anchor on AC?
10. Several studies on Ru SACs have reported the structure of Ru-N₄, but there are significant differences in catalytic activity and stability. What do the authors believe is the main reason for this?

Reviewer #3 (Remarks to the Author):

In this manuscript, the authors reported the acetylene hydrochlorination by asymmetric Ru-In atomic pairs with excellent yields and stability higher than 600 h. A deep study of the catalyst by experimental and theoretical techniques was performed to obtain that a d-p orbital interaction between Ru and In is the responsible of the selective absorption of acetylene in In and HCl in Ru, weaker than with pure Ru. This catalyst works better than the corresponding Ru₄/NC, that aggregates due to the over-chlorination.

However, in order to complete the study, is necessary a revision of the manuscript.

Catalyst characterization

It surprises me how the authors named the catalysts. For the monometallic ones (Ru and In) they denoted as RuN₄/NC and InN₄/NC, remarking that the metals are coordinated to 4 N from the support. However, the dimer is noted as RuIn/NC, avoiding that Ru and In are coordinated through a N since there is no evidence of Ru-In bond. If the active specie is Ru-N-In, why not name it with the N?

In figure 54 the authors have a picture that reveals the agglomeration of Ru in RuN₄/NC catalyst after reaction, but they do not show the same image for RuIn/NC catalyst, that is the important catalyst here, so they can not demonstrate that they do not observe agglomeration of the active species after reaction.

Catalytic Study

The authors achieve a deep characterization of the catalyst but regarding the catalytic study it seems that they only present the results without an analysis of them. For example, 180 °C reaction temperature is better than the higher ones (210 and 240 °C) What happens at these temperatures? Is it a problem of deabsorption of HCl or acetylene in the catalyst?

If the active specie is the asymmetric Ru-In atomic pairs, theoretically, the best ratio of starting materials should be Ru: In 1:1 in order to obtain a higher amount of dimers; however, the authors claimed that the best ratio Ru/In is 5, confirmed by ICP. There is an excess of Ru in the catalyst. Did you observe Ru single atoms in the sample or clusters? Since you claim in line 128 that a fraction of Ru and In appear in intimate atomic pairs, do you know the amount of Ru and In that are forming the dimers?

In Figure 54 the authors show the activity of other bimetallic catalyst obtained using the same procedure. Do the authors have any characterization data to confirm that they also obtain the asymmetric M-In atomic pairs?

The catalytic test was analysed by gas chromatography, but the authors forgot to describe the column used, necessary to reproduce the procedure.

The other products obtained with InN₄/NC are 1,1-dichloroethane and 1,2-dichloroethane. How they can confirm that these are the products obtained? No description facilitated in the manuscript or supporting.

References

In the introduction, page 2 line 24, the authors missed some references regarding the hydrochlorination of acetylene with Au, one of the most used catalyst to substitute Hg: J. Catal. 1985, 96, 292; Appl. Catal. 1985, 17, 155; J. Catal. 2007, 250, 231; Angew., 2017, 56, 6435.

Other issues

The graphs should be represented also with error bars.

Line 24: In recent, (in recent times, in recent years...?)

Line 65: polyaniline (PAN). Lines 365 and 368: PANI.

Scheme 1 and Figure 1: RuN₄/NC, InN₄/NC (subscript).

In supporting Figure 25, the scheme shows a bottle of H₂S not used in the experiment, or at least not mentioned in the manuscript. Please, delete to avoid misunderstandings.

Supporting Figure 35: The obtained data of the there (??) samples are the same as the reaction conditions of RuIn/NC.

Supporting Figure 46: please, indicate which colour is Ru and In.

Reference 29: paper volume in bold.

Response to the reviewers' comments:

Reviewer 1

To seek other metals to substitute the toxic mercuric chloride catalyst for acetylene hydrochlorination is of great importance. The Ru-based catalysts were reported to be active for this reaction and proposed to be potential candidates. In this manuscript, the authors reported a Ru single-atom catalyst (SAC) that can resist the aggregation of Ru atoms caused by over-chlorination by synthesizing the asymmetric Ru-In atomic pairs. Assisted by In, the loss of Ru could be greatly suppressed (from 4.27% to 0.20%) after reaction for 100 h. Besides, the stability of RuIn/NC catalyst was also greatly enhanced, which could run 600 h without obvious deactivation. This result is claimed to be the most outstanding Ru-based catalyst among all of those that were ever reported in the literatures. It can be accepted after carefully considering the following questions.

Response: Sincerely thank you for your rigorous and valuable suggestions. we have addressed each of these issues with additional experimental evidences, especially the characterization and analysis of the catalysts during and after the reaction. The quality of the revised manuscript has been improved with the help of editor and reviewers.

1. Although the activity and stability was enhanced by synthesizing RuIn/NC SAC, the evidences for the Ru-N-In structure were not solid enough. Firstly, because the atomic number of Ru and In was close to each other (44 vs 49), the HAADF-TEM results showed two atoms with similar lightness. Besides, only three pairs of Ru/In singles atoms were tested for their distances. This result could not stand for the whole sample. The existence of the Ru-N-Ru and In-N-In could not be excluded.

Response: Thank you for your rigorous and careful suggestion. The intensity of atoms in AC-HAADF-STEM image is proportional to the square of the atomic number. Although the atomic numbers of Ru and In are relatively close (44 vs 49), there is still a difference in the intensity of the two atoms in the AC-HAADF-STEM images.

In the original manuscript, we only marked three atomic Ru-In pairs and provided their distances in the ACHAADF-STEM image (**Figure 1d**). To strengthen the evidence about atomic distances between Ru-In sites confined by the carbon matrix, we have added the detailed intensity profile about AC-HAADF-STEM of Ru-N-In/NC to comprehensively summarize the distribution of Ru and In atoms as much as possible. As shown in **Figures R1~R2**, we have investigated total 19 of distances of Ru-In pairs decorated on the different surfaces of Ru-N-In/NC. As expected, all the results point to the same conclusion, that is, the as-prepared Ru-N-In/NC catalyst exhibits the Ru-In atomic pairs, and the average distance of the two neighboring Ru and In sites was determined as ~0.37 nm. Since all atomic pairs have exhibited different atomic intensities, it is reasonable to assume that the synthesized catalysts are heterogeneous Ru-N-In diatomic structures. The corresponding revisions have been added in **Manuscript** and **Supporting Information**. (**Lines 83~85**)

Indeed, as the reviewer notes, it is difficult to exclude the existence of the Ru-N-Ru and In-N-In on the basis of AC-HAADF-STEM alone. Thus, we simulated the structures of Ru-N-Ru, In-N-In, and Ru-N-In through density functional theory, and then compared their Gibbs free energy (**Figure R3**). The asymmetric Ru-N-In dual-atom structure has the lowest free energy of -2.793 eV, indicating that Ru-N-In dual-atom structure is the most stable configuration in comparison with the Ru-N-Ru and In-N-In. In addition, we applied the quantitative structural parameters analysis according to the fitted EXAFS spectra to further confirming the formation of Ru-N-In dual-atom is more favorable than two other possible dual-atom configurations of Ru-N-Ru and In-N-In. As shown in **Supplementary Table 4**, the too large ΔE_0 (eV) value (-60.4 ± 24.4) for Ru-N-Ru indicates that the Ru-N-Ru structure is irrational. The σ^2 (\AA^2) value (0.0172 ± 0.0262) for the In-N-In may be negative reveals that the In-N-In structure is also not reasonable. Furthermore, the corresponding EXAFS spectra cannot achieve the optimum fitting due to the both R factors are over 0.02.

Combining multiple characterizations and density functional theory mentioned in the above discussions, we tend to believe that the synthesized Ru-N-In/NC catalyst exhibits a Ru-N-In structure, in which Ru and In are anchored on NC structure through metal-N

bonding in coordination assemblies of Ru-N₃ and In-N₄ connected by N bridges.

Figure R1. AC-HAADF-STEM image of the Ru-N-In/NC sample, and the distance (nm) of isolated Ru and In dual single-atom sites from 4# to 11#.

Figure R2. AC-HAADF-STEM image of the Ru-N-In/NC sample, and the distance (nm) of isolated Ru and In dual single-atom sites from 12# to 19#.

Figure R3. Structure of Ru-N-Ru, In-N-In, and Ru-N-In models and their free energies. Orange, pink, blue, and gray spheres represent Ru, In, N, and C atoms, respectively.

Supplementary Table 4 EXAFS fitting parameters at the Ru and In *K*-edge for Ru-N-In/NC.

Sample	Coordination	CN	$R(\text{\AA})$	$\sigma^2(\text{\AA}^2)$	$\Delta E_0(\text{eV})$	R factor
Ru-N-In/NC	Ru-N	3.6±0.8	2.031±0.017	0.0086±0.0026	2.5±2.6	0.0211
	Ru-N-Ru	1.5±0.7	2.906±0.117	0.0109±0.0035	-60.4±24.4	
	In-N	4.0±0.7	2.147±0.014	0.0102±0.0026	2.2±1.7	0.0245
	In-N-In	2.0±0.8	3.558±0.036	0.0172±0.0262	4.1±9.1	

A reasonable range of EXAFS fitting parameters: $0.700 < S_0^2 < 1.000$; $CN > 0$; $\sigma^2 > 0 \text{ \AA}^2$; $|\Delta E_0| < 10 \text{ eV}$; R factor < 0.02 . Fitting range: $3.0 \leq k (\text{\AA}) \leq 13.5$ and $1.0 \leq R (\text{\AA}) \leq 3.0$ (Ru foil); $3.0 \leq k (\text{\AA}) \leq 10.5$ and $1.0 \leq R (\text{\AA}) \leq 2.5$ (Ru-N-In/NC); $3.0 \leq k (\text{\AA}) \leq 11.0$ and $2.0 \leq R (\text{\AA}) \leq 4.0$ (In foil); $3.0 \leq k (\text{\AA}) \leq 10.5$ and $1.0 \leq R (\text{\AA}) \leq 2.5$ (Ru-N-In/NC).

2. Secondly, the loss of In was not suppressed, which was 0.53% and 0.51% for the RuIn/NC and In/NC catalyst after reaction for 100 h, respectively. Was the loading of Ru and In tested after reaction for different time intervals (e.g., 100 h, 200 h, 300 h...) and compared to see if their loading became stable after a certain reaction time? If there were Ru-N-In interactions, would the loss of Ru and In be avoided?

Response: Thanks for your valuable suggestion. ICP-OES testing was implemented to track the metal content on the catalysts. Typically, the sample (10 mg) was added into a digestion tank containing 7 mL mixed acid solution ($V(\text{HCl}): V(\text{HNO}_3) = 2 : 1$) and 0.75 mL HF solution, and then digested and processed at 500 °C for several hours until the solid material was completely dissolved. Based on the above testing methods, we retested the contents of Ru and In after reaction for different time intervals from 0 to 600 h.

As seen in **Supplementary Table 1**, the loss amount (original load amount—residual load amount) of In element in Ru-N-In/NC (0.001 wt.%) was smaller than that in InN₄/NC (0.005 wt.%) after reaction for 100 hours, indicating that the loss of In was suppressed in

the Ru-N-In/NC catalyst. In addition, considering the potential misunderstanding caused by the inappropriate calculation method for loss rate (= loss amount/original load amount) used in the original manuscript, we have replaced the loss ratio by loss amount, as shown in the revised **Supplementary Table 1**. We have compared the loading of Ru and In contents after reaction for different time intervals from 0 h to 600 h. We observed that their loading became stable after the reaction time of 200 h. Thus, the loss of Ru and In can be effectively avoided due to the Ru-N-In interactions during the acetylene hydrochlorination reaction.

Supplementary Table 1 The fresh and used Ru/In loading, as well as the loss amount of Ru/In of Ru-N-In/NC, RuN₄/NC, and InN₄/NC catalysts, determined by ICP-OES.

Catalyst	Ru loading (wt.%)	Δ Ru (wt.%)	In loading (wt.%)	Δ In (wt.%)
Ru-N-In/NC-0 h	0.978	/	0.187	/
Ru-N-In/NC-100 h	0.972	0.006	0.186	0.001
Ru-N-In/NC-200 h	0.822	0.150	0.183	0.003
Ru-N-In/NC-300 h	0.820	0.002	0.182	0.001
Ru-N-In/NC-400 h	0.819	0.001	0.181	0.001
Ru-N-In/NC-500 h	0.819	0	0.180	0.001
Ru-N-In/NC-600 h	0.818	0.001	0.180	0
RuN ₄ /NC-0 h	0.984	/	/	/
RuN ₄ /NC-100 h	0.942	0.042	/	/
InN ₄ /NC-0 h	/	/	0.988	/
InN ₄ /NC-100 h	/	/	0.983	0.005

Δ Ru and Δ In represent the loss amount of Ru or In after the reaction for every 100 hours.

3. Thirdly, the EXAFS data could not tell the adjacent atom was In or Ru, either. In supplementary Table 4, the r -factor for the data fitting results was close to 0.02, besides, the other parameters were also reasonable. Therefore, there might be Ru-N-Ru/In (2.906 Å) and/or In-N-Ru/In (3.558 Å) interactions in the RuIn/NC sample. But it could not be the only evidence. The data fitting range of should be provided under supplementary Table 4.

Response: Thank for your careful check and kindly guidance. The data fitting range of **Supplementary Table 4** is shown as follows: $3.0 \leq k (\text{Å}^{-1}) \leq 13.5$ and $1.0 \leq R (\text{Å}) \leq 3.0$ (Ru foil); $3.0 \leq k (\text{Å}^{-1}) \leq 10.5$ and $1.0 \leq R (\text{Å}) \leq 2.5$ (Ru-N-In/NC); $3.0 \leq k (\text{Å}^{-1}) \leq 11.0$ and $2.0 \leq R (\text{Å}) \leq 4.0$ (In foil); $3.0 \leq k (\text{Å}^{-1}) \leq 10.5$ and $1.0 \leq R (\text{Å}) \leq 2.5$ (Ru-N-In/NC). The data fitting range of **Supplementary Table 4** is the same as **Supplementary Table 3**.

Actually, the parameters provided in **Supplementary Table 4** have some unreasonable aspects. Based on the fitting principles, a reasonable range of EXAFS fitting parameters is as follows: $0.700 < S_0^2 < 1.000$; $CN > 0$; $\sigma^2 > 0 \text{ Å}^2$; $|\Delta E_0| < 10 \text{ eV}$; $R \text{ factor} < 0.02$. (I) the value of ΔE_0 (eV) (-60.4 ± 24.4) for Ru-N-Ru is significantly too large, indicating that the Ru-N-Ru structure is irrational. (II) By considering the error, the σ^2 (Å^2) value of 0.0172 ± 0.0262 for the In-N-In structure may be negative. The σ^2 value should be positive according to the fitting principles, therefore a possible negative σ^2 value demonstrates that the In-N-In structure is also irrational. (III) Most importantly, the fitting data with R factor is more than 0.02 further demonstrates that the Ru-N-Ru and In-N-In structures are hard to form in our synthesized Ru-N-In/NC catalysts.^[R1-R3]

Supplementary Table 4 EXAFS fitting parameters at the Ru and In K -edge for Ru-N-In/NC.

Sample	Coordination	CN	$R(\text{Å})$	$\sigma^2(\text{Å}^2)$	$\Delta E_0(\text{eV})$	$R \text{ factor}$
Ru-N-In/NC	Ru-N	3.6 ± 0.8	2.031 ± 0.017	0.0086 ± 0.0026	2.5 ± 2.6	0.0211
	Ru-N-Ru	1.5 ± 0.7	2.906 ± 0.117	0.0109 ± 0.0035	-60.4 ± 24.4	
	In-N	4.0 ± 0.7	2.147 ± 0.014	0.0102 ± 0.0026	2.2 ± 1.7	0.0245
	In-N-In	2.0 ± 0.8	3.558 ± 0.036	0.0172 ± 0.0262	4.1 ± 9.1	

A reasonable range of EXAFS fitting parameters: $0.700 < S_0^2 < 1.000$; $CN > 0$; $\sigma^2 > 0 \text{ \AA}^2$; $|\Delta E_0| < 10 \text{ eV}$; $R \text{ factor} < 0.02$. Fitting range: $3.0 \leq k (\text{\AA}) \leq 13.5$ and $1.0 \leq R (\text{\AA}) \leq 3.0$ (Ru foil); $3.0 \leq k (\text{\AA}) \leq 10.5$ and $1.0 \leq R (\text{\AA}) \leq 2.5$ (Ru-N-In/NC); $3.0 \leq k (\text{\AA}) \leq 11.0$ and $2.0 \leq R (\text{\AA}) \leq 4.0$ (In foil); $3.0 \leq k (\text{\AA}) \leq 10.5$ and $1.0 \leq R (\text{\AA}) \leq 2.5$ (Ru-N-In/NC).

References

- [R1] Ravel, B. and Newville, M. ATHENA, ARTEMIS, HEPHAESTUS: data analysis for X-ray absorption spectroscopy using IFEFFIT. *J. Synchrotron Rad.* **12**, 537-541 (2005).
- [R2] Funke, H. et al. Wavelet Analysis of Extended X-ray Absorption Fine Structure Data. *Phys. Rev. B* **71**, 094110 (2005).
- [R3] Zabinsky, S. I. et al. Multiple-Scattering Calculations of X-Ray-Absorption Spectra. *Phys. Rev. B* **52**, 2995-3009 (1995).

4. The BET results in supplementary Table 2, there were too many significant figures. There were detection limits for the instrument, so they cannot be so precise.

Response: Thank you for your careful comment. We have corrected the figures in the revised **Supplementary Table 2**.

Supplementary Table 2 Pore structure parameters of the as-prepared Ru-based catalysts.

Catalyst	BET surface area (m ² /g)	Total pore volume (cm ³ /g)	Average pore diameter (nm)
NC600	802	0.37	2.68
NC700	930	0.40	2.52
NC800	1020	0.29	2.30
NC900	910	0.39	2.71
NC1000	753	0.34	3.39
RuN ₄ /NC800	678	0.35	3.69
InN ₄ /NC800	634	0.37	3.19
Ru-N-In/NC800	690	0.43	2.72

5. In supplementary Table 7, why there were two r -factors for one group of data (Ru-N and Ru-Cl)?

Response: We apologize for this mistake and thank the reviewer very much. One of the r -factor was the typo. We have revised **Supplementary Table 7**.

Supplementary Table 7 EXAFS fitting parameters of the Ru K -edge for various post-hydrochlorination samples.

Samples	Coordination	CN^a	$R(\text{\AA})^b$	$\sigma^2(\text{\AA}^2)^c$	$\Delta E_0(\text{eV})^d$	R factor
Ru-N-In/NC-3 h	Ru-N	3.2±0.2	1.966±0.003	0.0049±0.0012	5.2±1.3	0.0038
	Ru-Cl	0.9±0.3	2.391±0.014	0.0090±0.0017	3.3±2.1	
RuN ₄ /NC-1 h	Ru-N	3.1±0.3	2.012±0.008	0.0035±0.0009	1.5±2.3	0.0095
	Ru-Cl	1.3±0.2	2.364±0.021	0.0025±0.0021	3.2±1.6	
RuN ₄ /NC-3 h	Ru-N	3.2±0.4	2.029±0.005	0.0091±0.0015	4.6±2.8	0.0089
	Ru-Cl	2.6±0.5	2.378±0.011	0.0043±0.0022	1.2±1.6	

^a CN , coordination number; ^b R , the distance to the neighboring atom; ^c σ^2 , the mean square relative displacement (MSRD); ^d ΔE_0 , inner potential correction; R factor indicates the goodness of the fit. A reasonable range of EXAFS fitting parameters: $0.700 < S_0^2 < 1.000$; $CN > 0$; $\sigma^2 > 0 \text{ \AA}^2$; $|\Delta E_0| < 10 \text{ eV}$; $R \text{ factor} < 0.02$.

6. In supplementary Table 8, the ratio of Ru-Cl of the RuN₄/NC sample decreased with reaction time, while that of the RuIn/NC sample became almost stable after reaction for 1 h. What is the possible reason?

Response: Thanks for your important comment. In recent years, Ru single-atom catalysts (especially Ru-N₄ configuration) have emerged as the promising candidates due to their flexible control of architectures. However, Ru single-atom catalysts always suffer from the easy deactivation caused by the over-chlorination of Ru active centers. This is the most important result of our current study (**Figure R4** and **Scheme 1**).

Over-chlorination process

Figure R4. The over-chlorination process of the symmetrical RuN₄/NC moiety during the hydrochlorination reaction.

According to the mechanisms proposed in this study, HCl not only acts as a reactant, but also serves as a chlorinating agent during acetylene hydrochlorination reaction. Briefly, the intrinsic properties of Ru in RuN₄/NC such as high cohesive energy (~8.0 eV) and Lewis acidity, readily promoting the excessive adsorption of Cl* (Lewis base) on the surface of catalysts. While the chlorinated Ru atoms tend to be migrated when exposed to HCl flow due to its smaller dissociation energy of Ru (1.27 eV of RuN₄/NC vs 1.86 eV of Ru-N-In/NC) (**Figure 5a**). Thus, the migration of Ru results in the decreased content of Ru-Cl species for RuN₄/NC samples. Since our reaction was carried out under continuous flow conditions, a decrease in the Ru-Cl content would result in the increased deposition of Cl on the carbon carriers to form the C-Cl species.^[R1~R4]

In contrast, the results of XPS confirm the Ru-N-In/NC catalyst can effectively resist the over-chlorination, as the content of Ru-Cl became almost stable after reaction for one hour (**Supplementary Figure 58**). Generally speaking, the constructed asymmetric Ru-N-In/NC catalysts can address the issue of the migration of Ru induced by the over-chlorination. The introduction of In atoms allows for the selective adsorption of C_2H_2 and HCl on each metal site respectively and mediates the steric hindrance happened when two reactant molecules adsorbed on atomic Ru site in the case of RuN_4/NC , thus enhancing the performances of acetylene hydrochlorination. Furthermore, a $d-p$ orbital coupling between Ru-In atom pairs modulated the electron configuration of Ru, resulting in the interaction of p orbital of Cl with $d_{x^2-y^2}$ orbital of Ru, rather than the d_{yz} and d_{xz} in the case of the RuN_4/NC catalyst. These results contributed a higher energy barrier for the over-chlorination process, and thermodynamic transition from exothermal to endothermal, which inhibited the migration of Ru atoms, and then dramatically improved the stability for acetylene hydrochlorination.

Figure 5a. Ru dissociation energy of Ru-N-In/NC and RuN_4/NC moieties.

Supplementary Figure 58. Cl 2p XPS spectra of the fresh and progressively chlorinated (a) Ru-N-In/NC and (b) RuN₄/NC samples.

References

- [R1] Kaiser, S. K. et al. Design of Carbon Supports for Metal-Catalyzed Acetylene Hydrochlorination. *Nat. Commun.* **12**, 4016 (2021).
- [R2] Kaiser, S. et al. Performance Descriptors of Nanostructured Metal Catalysts for Acetylene Hydrochlorination. *Nat. Nanotechnol.* **17**, 606-612 (2022).
- [R3] Peng, J. et al. Manipulating Micro-Electric Field and Coordination-Saturated Site Configuration Boosted Activity and Safety of Frustrated Single-Atom Cu/O Lewis Pair for Acetylene Hydrochlorination. *Nano Res.* **16**, 6178-6186 (2023).
- [R4] Giulimondi, V. et al. Evidence of Bifunctionality of Carbons and Metal Atoms in Catalyzed Acetylene Hydrochlorination. *Nat. Commun.* **14**, 5557 (2023).

Reviewer 2

This manuscript by Fan *et al.* described a Ru-In dual single-atom catalyst, possessing impressive catalytic stability for acetylene hydrochlorination. Abundant characterizations and theoretical calculations were conducted to reveal the superior catalytic performance and several interesting arguments were proposed, which might arouse interests for readers and provide insights for future SACs design. Some major concerns should be addressed though before I can recommend for the publication on *Nat. Commun.*

Response: Sincerely thanks for your meticulous review and constructive criticisms. We have re-conducted the related simulation calculations and provided detailed analysis to address the issues one by one in response to your comments. Further, more experimental and characterization evidences have also been provided to support our results. The quality of the revised manuscript has been improved with the help of editor and reviewers.

1. As is well known, the intensity of the image in STEM is proportional to the square of the atomic number. However, in Figure 1e, f and g, the intensity of Ru (44 #) atom is higher than that of In (49 #), please explain. Moreover, I am concerned whether STEM can distinguish these two elements to obtain the conclusion of dual Ru-In SAC.

Response: Thank you for your comment. We are very sorry for the misunderstanding of the atomic strength of Ru and In in the ACHAADF-STEM image shown in **Figures 1e, f, and g**. We have corrected this mistake in the revised **Figures 1e, f, and g**, and then revised the corresponding descriptions. **(Lines 83~85)**

As mentioned by the reviewer, STEM cannot accurately distinguish the two elements to obtain the conclusion of dual Ru-In atomic pairs. Therefore, our conclusion about Ru-In dual-atom pairs is not simply according to the results of the ACHAADF-STEM images, but rather on the results of multiple characterizations, including electron microscope, X-ray absorption spectroscopy, and density functional theory. In the original manuscript, we only marked three atomic Ru-In pairs and provided their distances in the ACHAADF-STEM

image (Figure 1d). To further distinguish the Ru-In atomic pairs and their atom distances, we have added additional atomic intensity profiles about AC-HAADF-STEM of the Ru-N-In/NC catalyst to comprehensively summarize the distribution of Ru and In atoms as much as possible. As presented in Figures R1~R2, we have investigated total 19 of distances of Ru-In pairs decorated on the different surface of Ru-N-In/NC. All atomic pairs exhibited different atomic intensities, we therefore highly speculated that the synthesized Ru-N-In/NC catalyst is a heterogeneous Ru-In diatomic structure.

Figure R1. AC-HAADF-STEM image of the Ru-N-In/NC sample, and the distance (nm) of isolated Ru and In dual single-atom sites from 4# to 11#.

Figure R2. AC-HAADF-STEM image of the Ru-N-In/NC sample, and the distance (nm) of isolated Ru and In dual single-atom sites from 12# to 19#.

Subsequently, the identification of single-atom states, coordination environments, and local structures of Ru and In atoms were determined by X-ray absorption spectroscopy (**Figure 2**). XANES and wavelet transform plots indicated that Ru and In are atomically dispersed on the NC supports, in which Ru and In sites appearing in adjacent atomic pairs. Moreover, EXAFS characterization and corresponding quantitative structural parameters analysis indicated that each Ru atom coordinates with three N atoms in the inter-plane of the tri-striazine framework with bond length of ≈ 1.99 Å, and each In atom is coordinated with four N atoms with an average In-N bond length of ≈ 2.15 Å. In addition, to verify the accuracy of Ru-N-In structure, we also considered two possible dual-atom configurations (e.g., Ru-N-Ru and In-N-In) similar to Ru-N-In, but the corresponding EXAFS spectra (R factor > 0.02) suggested these structures were less possible to be obtained in comparison with the Ru-N-In configuration (**Supplementary Tables 3~4**).

Figure 2. (a) Normalized XANES, (b) Fourier-transform XAFS spectra, and (c) XAFS curves fitting in R space. In K-edge XAFS analysis of Ru-N-In/NC: (d) Normalized XANES, (e) Fourier transform XAFS spectra, and (f) EXAFS curves fitting in R space. (g) Wavelet transform contour plots of Ru-N-In/NC at Ru K-edge and at In K-edge.

Eventually, we compared the thermodynamic formation energies for six single-atom or dimer-atom structures, including Ru-N-Ru/NC, In-N-In/NC, Ru-N-In/NC, RuN₃/NC, RuN₄/NC, and InN₄/NC, which their formation energies are thermodynamically favorable. Especially, asymmetric Rn-In dual-atom structure was identified as the most favorable configuration that has the lowest free energy of -2.793 eV (**Supplementary Figure 9**). In summary, the coordination environment of Ru-N-In has been unveiled based the above

multiple characterizations, in which Ru and In are anchored on NC structure through metal-N bonding in coordination assemblies of Ru-N₃ and In-N₄ connected by N bridges.

Supplementary Figure 9. Structure of various Ru-based models and their free energies calculated by DFT. Orange, pink, blue, and gray spheres represent Ru, In, N, and C atoms, respectively.

2. The result shows that the optimal wt. or mol. Ru/In ratio is ~ 5:1 (**Figure S27**). If the Ru-In dual single-atom site is proposed to be the most active, why is the optimal Ru/In ratio not close to 1?

Response: Thanks for your rigorous and important comments. As for the optimal Ru/In weight ratio of 5:1 (i.e., ~5 as mol ratio), we believe that more Ru-N-In atom pairs can be generated in this case, since there are indeed some single atom sites can be identified in STEM image of Ru-N-In/NC catalyst (**Figure 1**). To further prove this point of the generation of more Ru-N-In atom pairs, the DRIFTS for CO adsorption studies were conducted to quantify different Ru active centers (Ru₁-N₄ and Ru-N-In)

As illustrated in **Figure R5**, the band at 2129 and 2121 cm⁻¹ could be attributed to the multicarbonyl species adsorbed on partially oxidized Ru^{δ+} sites (donated as Ru^{δ+}(CO)_x species with x = 2 or 3). Compared with the Ru-N₄ sample, the peak at 2121 cm⁻¹ can be

attributed to the interaction between CO and Ru-N-In sites, while the peak at 2129 cm^{-1} is assigned to the interaction between Ru-N₄ and CO. The broad band locates in the range of $2000\sim 1920\text{ cm}^{-1}$, and is centered at around 1971 cm^{-1} , which could be ascribed to the linear-adsorbed CO on the boundary between Ru sites and the support or bridge-adsorbed CO on Ru sites.^[R1-R5]

Figure R5. In situ DRIFTS spectra and the peak deconvolution analysis of CO adsorption on the Ru-N-In/NC and RuN₄/NC catalysts at 180 °C.

Figure R6. Ru-N-In content as a function of the Ru/In ratio, and the initial acetylene

conversion efficiency as a function of the Ru-N-In species content.

Table R1 Fitting parameters derived from In situ DRIFTS spectra of CO adsorption of the different Ru-N-In/NC and Ru/NC catalysts.

Catalysts	Ru-N-In		Ru-N ₄	
	Position (cm ⁻¹)	Content (%)	Position (cm ⁻¹)	Content (%)
Ru-N-In/NC Ru : In = 10 : 1		63.3		36.7
Ru-N-In/NC Ru : In = 5 : 1		79.6		20.4
Ru-N-In/NC Ru : In = 2 : 1	2121	58.9	2129	41.1
Ru-N-In/NC Ru : In = 1 : 1		53.8		46.2
RuN ₄ /NC	/	/		100

Subsequently, we performed peak deconvolution analysis for the DRIFTS spectra of CO adsorption to further quantify different active centers (Ru-N₄ and Ru-N-In). As shown in **Figure R6** and **Table R1**, a volcano-type correlation was observed between the Ru-N-In content and Ru/In ratio, confirming the highest level of Ru-N-In content (79.6%) was achieved in the case of Ru/In ratio of ~5:1. Furthermore, the initial acetylene conversion efficiency and Ru-N-In content follows a linear correlation ($R^2 = 0.94$), where the highest catalytic activity was obtained at the highest levels of Ru-N-In species. Only ~50% of the Ru atoms in the Ru-N-In/NC catalysts synthesized by our methods are in the Ru-N₄ form at the Ru/In ratio of 1. In this case, too much Ru-N₄ species anchored on the surface of Ru-N-In/NC is not beneficial for enhancing the activity and stability due to the inevitable influence of over-chlorination. Moreover, the chlorinated Ru-N₄ also triggers the migration and loss of Ru atoms. In contrast, the more Ru-N-In species can perfectly address this issue to sustain a superior performance.

In conclusion, based on current experimental and characterization results, the optimal Ru/In ratio was identified as ~ 5:1 to obtain more Ru-N-In atom pairs. In future studies, our research team will further involve in increasing the content of atom pairs to achieve the better performance in acetylene hydrochlorination.

References

- [R1] Lyu, S. et al. Dopamine Sacrificial Coating Strategy Driving Formation of Highly Active Surface-Exposed Ru Sites on Ru/TiO₂ Catalysts in Fischer-Tropsch Synthesis. *Appl. Catal. B-Environ.* **278**, 119261 (2020).
- [R2] Abdel-Mageed, A. M. et al. Selective CO Methanation on Ru/TiO₂ catalysts: Role and Influence of Metal-Support Interactions. *ACS Catal.* **5**, 6753-6763 (2015)
- [R3] Panagiotopoulou, P. et al. Mechanistic Study of the Selective Methanation of CO over Ru/TiO₂ Catalyst: Identification of Active Surface Species and Reaction Pathways. *J. Phys. Chem. C* **115**, 1220-1230 (2011).
- [R4] Elmasides, C. et al. Partial Oxidation of Methane to Synthesis Gas over Ru/TiO₂ Catalysts: Effects of Modification of the Support on Oxidation State and Catalytic Performance. *J. Catal.* **198**, 195-207 (2001).
- [R5] Davydov, A. A. and Bell, A. T. An Infrared Study of NO and CO Adsorption on a Silica-Supported Ru Catalyst. *J. Catal.* **49**, 332-344 (1977).

3. The characterizations in the article are based on catalysts with a Ru/In molar ratio of ~ 5:1. Then the authors reported the structure of dual Ru-In single atom site with a Ru-N coordination number of 3.3. What is the coordination structure of excess Ru atoms without the pair with In? According to the structure of RuN₄ in Ru/NC, the Ru-N coordination number in RuIn/NC should be closer to 4 rather than 3 If N₄ coordination is adopted (Excess Ru atoms are more than the Ru paired with In).

Response: Thanks for your crucial suggestion. We fully agree with your viewpoint that Ru single atoms are unavoidably produced during the synthesis process. According to the quantitative results obtained from the DRIFTS spectra of CO adsorption, the Ru-N-In/NC

catalyst contains 79.6% Ru-In atomic pairs and 20.4% Ru single atoms. This 20.4% of Ru single atoms show the same adsorption characteristics as for RuN₄/NC (**Figure R7**). Thus, we considered that the coordination structure of excess Ru atoms without pairing with In atoms should be Ru-N₄ structure.

Figure R7. The peak deconvolution analysis of DRIFTS spectra for CO adsorption on the Ru-N-In/NC and RuN₄/NC catalysts at 180 °C.

Since we did not take into account the coexistence of Ru-In atomic pairs and Ru single atoms when we initially did the EXAFS quantitative structural parameters analysis in the original manuscript, and the fitting resulted in a coordination number for Ru-N close to 3.3. Hence, we again fitted the original XAS data when considering the co-existence of Ru-In atomic pairs and Ru single atoms, and then we obtained the Ru-N coordination number of 3.5. As the reviewer mentioned, compared with the original coordination number of 3.3, the updated coordination number of the Ru-N in Ru-N-In/NC is closer to 4. We have revised the original fitting data in **Supplementary Table 3**. Since the fitting of XAS data can only obtain an average value of coordination number, the coordination number of Ru-N in our original proposed Ru-In diatomic structure should remain 3. The increase of

Ru coordination number in the revised **Supplementary Table 3** is only due to the presence of the Ru-N₄ single atoms, which raises this average coordination number to 3.5.

Supplementary Table 3 EXAFS fitting parameters at the Ru and In *K*-edge for various samples.

Samples	Coordination	CN^a	$R(\text{\AA})^b$	$\sigma^2(\text{\AA}^2)^c$	$\Delta E_0(\text{eV})^d$	R factor
Ru foil	Ru-Ru	12*	2.672±0.002	0.0040±0.0005	-3.2±0.8	0.0038
Ru-N-In/NC	Ru-N	3.5±0.3	1.989±0.012	0.0101±0.0012	-1.5±2.0	0.0043
RuN₄/NC	Ru-N	3.9±0.3	2.042±0.004	0.0034±0.0011	-2.0±1.2	0.0062
In foil	In-In	12*	3.249±0.021	0.0241±0.0023	1.8±0.5	0.0074
Ru-N- In /NC	In-N	4.1±0.3	2.134±0.005	0.0112±0.0016	2.4±1.5	0.0102

^a*CN*, coordination number; ^b*R*, the distance to the neighboring atom; ^c σ^2 , the mean square relative displacement (MSRD); ^d ΔE_0 , inner potential correction; *R* factor indicates the goodness of the fit. S_0^2 was fixed to 0.863 and 0.788, according to the experimental EXAFS fit of Ru foil and In foil by fixing *CN* as the known crystallographic value. Fitting range: $3.0 \leq k (\text{\AA}^{-1}) \leq 13.5$ and $1.0 \leq R (\text{\AA}) \leq 3.0$ (Ru foil); $3.0 \leq k (\text{\AA}^{-1}) \leq 10.5$ and $1.0 \leq R (\text{\AA}) \leq 2.5$ (**Ru-N-In/NC**); $3.0 \leq k (\text{\AA}^{-1}) \leq 12.5$ and $1.0 \leq R (\text{\AA}) \leq 2.5$ (**RuN₄/NC**); $3.0 \leq k (\text{\AA}^{-1}) \leq 11.0$ and $2.0 \leq R (\text{\AA}) \leq 4.0$ (In foil); $3.0 \leq k (\text{\AA}^{-1}) \leq 10.5$ and $1.0 \leq R (\text{\AA}) \leq 2.5$ (Ru-N-**In**/NC). A reasonable range of EXAFS fitting parameters: $0.700 < S_0^2 < 1.000$; $CN > 0$; $\sigma^2 > 0 \text{\AA}^2$; $|\Delta E_0| < 10 \text{ eV}$; *R* factor < 0.02 .

4. Followed by the previous one, RuIn/NC exhibited much greater stability than Ru/NC. But how to explain the stability of excess Ru without pairing with In?

Response: Thanks for your valuable comment. Based on the **Comment 3**, the coordination structure of excess Ru without pairing with In atoms should be Ru-N₄ structure, which are inevitably generated during the synthesis of the Ru-N-In/NC catalysts. Obviously, such a Ru single-atom structure can exert an impact on the overall stability of the catalysts as indicated by the reviewer and several publications.^[R1~R4] However, based on the results of

the DRIFTS spectra for the Ru-N-In/NC and RuN₄/NC catalysts (Comment 2, **Figure R5**), we clearly observed that the Ru-N-In moiety is a dominated structure over Ru single atoms, as the proportion of Ru-In diatoms is 79.6%, close to 80%, while Ru single atoms are only 20.4%. Thus, we can conclude that the Ru-N-In/NC catalyst presents a stability of up to 600 hours is mainly provided by the presence of Ru-In atom pairs.

Figure 3c. Long-term catalytic performances of Ru-N-In/NC and RuN₄/NC. [Reaction conditions: T = 180 °C, $V_{\text{cat.}} = 1.2$ mL, $V(\text{HCl})/V(\text{C}_2\text{H}_2) = 1.15$, $GHSV(\text{C}_2\text{H}_2) = 180$ h⁻¹]

To validate the above conclusion, we firstly confirmed the stability of RuN₄/NC, which can only achieve a 120-hour running (**Figure 3c**). Subsequently, we have compared the loading of Ru and In contents after reaction for different time intervals from 0 h to 600 h through ICP-OES (**Supplementary Table 1**). We observed that their loading became stable after the reaction time of 200 h, indicating the loss of Ru and In can be effectively avoided due to the interaction of Ru-In atomic pairs during the acetylene hydrochlorination reaction. Furthermore, we calculated the Ru loss ratio of ~16% after reaction of 200 hours, close to 20% of Ru single atoms determined by the DRIFTS spectra of CO adsorption. It can be concluded that the Ru species without pairing with In atoms have already been lost in the early stages of catalytic reactions with Ru-N-In/NC.

In addition, we further compared the coke deposition of the Ru-N-In/NC and RuN₄/NC catalysts after acetylene hydrochlorination reaction (**Figure 5e**). It can be observed that a

slight coke deposition (ca. 0.18%) covered on the surface of Ru-N-In/NC, while a high coke coverage (ca. 5.23%) was accumulated over the surface of RuN₄/NC. We can attribute this slight carbon deposition to the presence of Ru-N₄ moieties without pairing with In atoms. Since the accumulation of coke on the Ru single atoms can directly lead to the deactivation of catalysts,^[R5~R7] the stability of RuN₄/NC was only 120 hours. In contrast, the construction of Ru-In atomic pairs by introducing the In atoms can effectively reduce the generation of coke accumulation, which provides a solid guarantee for long-term operation of 600 h.

Overall, combining stability experiments and characterization data, it can be concluded that the strong stability of Ru-N-In/NC comes from the contribution of Ru-In pairs, not the Ru species without pairing with In atoms. The presence of excess Ru single atoms without pairing with In atoms has no significant effect on the stability of Ru-N-In/NC catalyst. Increasing the content of atomic pairs in the diatomic catalysts to achieve higher catalytic performances is still a hot and difficult issue, which we will continue to investigate in our future research.

Supplementary Table 1 The fresh and used Ru/In loading, as well as the loss amount of Ru/In of Ru-N-In/NC catalysts, determined by ICP-OES.

Catalyst	Ru loading (wt.%)	Δ Ru (wt.%)	In loading (wt.%)	Δ In (wt.%)
Ru-N-In/NC-0 h	0.978	/	0.187	/
Ru-N-In/NC-100 h	0.972	0.006	0.186	0.001
Ru-N-In/NC-200 h	0.822	0.150	0.183	0.003
Ru-N-In/NC-300 h	0.820	0.002	0.182	0.001
Ru-N-In/NC-400 h	0.819	0.001	0.181	0.001
Ru-N-In/NC-500 h	0.819	0	0.180	0.001
Ru-N-In/NC-600 h	0.818	0.001	0.180	0

Δ Ru and Δ In represent the loss amount of Ru or In every 100 hours.

Figure 5e. Coke deposition of the post-hydrochlorination Ru-N-In/NC and RuN₄/NC.

References

- [R1] Song, Y. et al. Surface Activation by Single Ru Atoms for Enhanced High-Temperature CO₂ Electrolysis. *Angew. Chem. Int. Ed.* **63**, e2023133 (2024).
- [R2] Giulimondi, V. et al. Evidence of Bifunctionality of Carbons and Metal Atoms in Catalyzed Acetylene Hydrochlorination. *Nat. Commun.* **14**, 5557 (2023).
- [R3] Yao, Y. et al. Single Atom Ru Monolithic Electrode for Efficient Chlorine Evolution and Nitrate Reduction. *Angew. Chem. Int. Ed.* **61**, e202208215 (2022).
- [R4] Li, Z. et al. Well-Defined Materials for Heterogeneous Catalysis: From Nanoparticles to Isolated Single-Atom Sites. *Chem. Rev.* **120**, 623-682 (2020).
- [R5] Liu, L. and Corma, A. Metal Catalysts for Heterogeneous Catalysis: From Single Atoms to Nanoclusters and Nanoparticles. *Chem. Rev.* **118**, 4981-5079 (2018).
- [R6] Martín, A. J. et al. Unifying Views on Catalyst Deactivation. *Nat. Catal.* **5**, 854–866 (2022).
- [R7] Scott, S. L. A Matter of Life (time) and Death. *ACS Catal.* **8**, 8597-8599 (2018).

5. The optimal adsorption mode of acetylene on RuIn/NC showed by DFT (**Figure S46**) seems to be inconsistent with the results of in-situ DRIFTS, and no chemisorption characteristics were observed by DFT. In addition, it is suggested the author consider the adsorption free energy of acetylene at the experimental temperature, which could be significantly different from the electronic energy.

Response: Thank you for pointing out this issue. The results of *in-situ* DRIFTS shows that the $\nu_{\text{as}}(\text{C}_2\text{H}_2)$ of 3258 cm^{-1} band underwent a downward shift with reference to the gas-phase value at 3287 cm^{-1} , indicating that the C_2H_2 molecule was subjected to a strong chemisorption of In-CHCH due to its significant bond polarization (**Figure 4b**).^[R1~R3] Following your valuable suggestions, we have reconstructed these adsorption models by considering the adsorption free energy of acetylene at the experimental temperature of $180\text{ }^\circ\text{C}$. The calculation methods of Gibbs free energy change are shown as follows. The chemisorption characteristics have also illustrated in the revised **Supplementary Fig. 49**.

The calculation of adsorption Gibbs free energy is defined as:

$$\Delta G = \Delta E + \Delta ZPE - T\Delta S \quad (\text{R1})$$

where ΔE is the electronic energy calculated with VASP, ΔZPE and ΔS are the zero-point energy difference and the entropy change between the products and reactants, respectively, and T is the temperature (453.15 K).

Supplementary Figure 49. Screening of the adsorption structure of the C_2H_2 molecule on Ru-N-In/NC site. The first is the optimal structure. White, gray, blue, orange, and pink balls represent H, C, N, Ru, and In atoms, respectively.

References

- [R1] Albani, D. et al. Semihydrogenation of Acetylene on Indium Oxide: Proposed Single-Ensemble Catalysis. *Angew. Chem. Int. Ed.* **56**, 10755-10760 (2017).
- [R2] Song, Q. et al. Electrostatic Force-Driven Lattice Water Bridging to Stabilize a Partially Charged Indium MOF for Efficient Separation of C_2H_2/CO_2 mixtures. *J. Mater. Chem. A* **10**, 9363-9369 (2022).
- [R3] Cao, Y. et al. Selective Hydrogenation of Acetylene over Pd-In/ Al_2O_3 Catalyst: Promotional Effect of Indium and Composition-Dependent Performance. *ACS Catal.* **7**, 7835-7846 (2017).

6. In the calculated reaction pathway (**Figure 4g**), the co-adsorption configuration of C₂H₂ and HCl on Ru-In site appears unusual and inconsistent with the separately optimized configurations (**Figure S46 and S49**). Is this the most stable co-adsorption configuration?

Response: Thank you for your careful checking. According to the **Comment 5** mentioned by the reviewer, we did not consider the adsorption free energy at the experimental temperature (180 °C) when constructing the separation adsorption models of C₂H₂ and HCl, thus, this resulted in the co-adsorption configurations of C₂H₂ and HCl on the Ru-In sites being inconsistent with the separate configurations in original manuscript and supporting information. Following your suggestions, we have reconstructed the separate adsorption models, and then obtained the adsorption energies of C₂H₂ (**Supplementary Figure 49, Lines 219~221**) and HCl (**Supplementary Figure 52, Lines 230~232**) at the experimental temperature of 180 °C, furthermore, the chemisorption characteristics have also illustrated in the revised figures. In addition, we compared four co-adsorption configurations (**Figure R8**) and their Gibbs free energies. The first one configuration displayed in **Figure R8** and **Figure 4g** is the most stable co-adsorption configuration, and this structure is consistent with the separately optimized configurations towards C₂H₂ and HCl. Compared with other co-adsorption configurations, the first one configuration displayed in **Figure R8** indicates that the C₂H₂ molecule is subjected to a strong chemisorption with In atom via *p*- π bonding, while the HCl molecule linearly adsorbs on atomic Ru site, the separated adsorption configuration on Ru-N-In atomic pairs facilitates the next hydrogenation step.

Figure R8. The co-adsorption structures of the C_2H_2 and HCl molecules on Ru-N-In/NC site. The first is the optimal structure. White, gray, blue, green, orange, and pink balls represent H, C, N, Cl, Ru, and In atoms, respectively.

Supplementary Figure 52. Screening of the adsorption structure of the HCl molecule on Ru-N-In/NC site. The first is the optimal structure. White, gray, blue, green, orange, and pink balls represent H, C, N, Cl, Ru, and In atoms, respectively.

7. The model in Figure 5d seems to have some issues, as N atoms do not replace C but instead appear above C atoms.

Response: Thanks for your careful comment. We have revised the **Figure 5d** by replacing carbon atoms with nitrogen atoms. The updated **Figure 5d** is as follows.

Original version

Revised version

Figure 5d. Scheme of the evolution pathway of Ru species over the Ru-N-In/NC and RuN₄/NC configurations. Gray, pink, purple, yellow and green balls represent C, N, Ru, In, and Cl atoms, respectively.

8. The author proposes a mechanism by which the chlorination of Ru single atoms leads to its migration out of the N₄ site. This is interesting. But I wonder what is the “chlorinated atmosphere” referred to by the authors in the manuscript and the caption of **Figure S62**, and is it HCl? If not please add an explanation in the catalytic evaluation section. Hydrogen chloride is not commonly considered as a reducing gas, why can RuN₄ be reduced to Ru by HCl, and what is the potential mechanism?

Response: Thanks for your constructive comment. The “chlorinated atmosphere” here is not just HCl, but the chlorine-containing reaction atmosphere of C₂H₂ + HCl (both their purities are over 99 %, $V(\text{HCl})/V(\text{C}_2\text{H}_2) = 1.15$). To avoid misunderstanding, we have added the explanation in the caption of **Supplementary Figure 65**.

The revised version: Supplementary Figure 65. The Ru 3*p* XPS spectra of the RuN₄/NC catalyst after treating by the reactive atmosphere. The reactive atmosphere here refers to the atmosphere of C₂H₂ + HCl ($V(\text{HCl})/V(\text{C}_2\text{H}_2) = 1.15$) for acetylene hydrochlorination, in which the gas purities are over 99 %.

Lines 330–333: Unfortunately, the active Ru species of RuN₄/NC in comparison with Ru-N-In/NC was completely deactivated to the Ru₀ under the reactive atmosphere (C₂H₂ + HCl), as demonstrated by Ru 3*p* XPS.

The reasons for the active Ru species in RuN₄ can be reduced to Ru(0) are as follows: (i) Several recent studies have demonstrated that an important reason for the reduction of active Ru(III) or Ru(IV) species to inactive Ru(0) can be attributed to the thermodynamically favored auto-reduction and/or in-excess reductants (reductive impurities, such as C₂H₂). At the reaction temperatures of 120~180 °C, Ru(III)/Ru(IV) can be partially reduced under the influence of thermal conditions. As long as Ru(0) or Ru particle is formed, the catalyst would suffer from irreversible and aggregation on surface leading to deactivation.^[R1~R4]

Supplementary Figure 55. NH₃-IR profiles of Ru-N-In/NC, RuIn/AC, RuN₄/NC, and InN₄/NC catalysts.

Figure 5a. Ru dissociation energy of Ru-N-In/NC and RuN₄/NC moieties.

(ii) According to the mechanism proposed in this study, HCl, which is not commonly considered as a reducing gas, does not act as a reducing agent for metal atoms, but acts as a chlorinating agent. This is the most important finding in this investigation. Briefly, the intrinsic properties of Ru (Lewis acid) in RuN₄ structure such as high cohesive energy (~8.0 eV) and Lewis acidity (**Supplementary Figure 55**), promoting the excessive adsorption of Cl* (Lewis base) species on catalyst surfaces, which becomes energy-favorable thermodynamically spontaneous process according to our DFT calculation results (**Figure 5d**), resulting in that the chlorinated Ru atoms tend to be migrated when exposed to HCl flow due to its weakened dissociation energy (**Figure 5a**). The remaining Ru atoms that have not been migrated by Cl* gradually aggregate due to the breakage of structure. Eventually, the active Ru atoms are reduced to inactive Ru(0) under the C₂H₂ atmosphere.^[R5~R7]

Overall, the reduction of reactive Ru in the RuN₄ configuration is mainly caused by the strongly reducing acetylene atmosphere in addition to thermodynamic factors. HCl is an important triggering factor for the reduction and deactivation of Ru.

References

[R1] Ye, L. et al. Self-Regeneration of Au/CeO₂ Based Catalysts with Enhanced Activity and Ultra-stability for Acetylene Hydrochlorination. *Nat. Commun.* **10**, 914 (2019).

[R2] Kaiser, S. et al. Preserved in a Shell: High-Performance Graphene-Confined Ruthenium Nanoparticles in Acetylene Hydrochlorination. *Angew. Chem. Int. Ed.* **58**, 12297-12304 (2019).

[R3] Kaiser, S. K. et al. Design of Carbon Supports for Metal-Catalyzed Acetylene Hydrochlorination. *Nat. Commun.* **12**, 4016 (2021).

[R4] Martín, A. J. et al. Unifying Views on Catalyst Deactivation. *Nat. Catal.* **5**, 854-866 (2022).

[R5] Kaiser, S. et al. Performance Descriptors of Nanostructured Metal Catalysts for Acetylene Hydrochlorination. *Nat. Nanotechnol.* **17**, 606-612 (2022).

[R6] Wang, X. et al. Progress of *p*-Block Element-Regulated Catalysts for Acetylene

Hydrochlorination. *Coordin. Chem. Rev.* 500, 215541 (2024).

[R7] Giulimondi, V. et al. Evidence of Bifunctionality of Carbons and Metal Atoms in Catalyzed Acetylene Hydrochlorination. *Nat. Commun.* **14**, 5557 (2023).

9. Note that RuIn/AC without N also has good catalytic performance and inhibition of coke deposition. Do Ru and In also exhibit single atom dispersion? How do they anchor on AC?

Response: Thanks for your important comment. To avoid misunderstandings, we provide more details of the synthesis process of RuIn/AC (Lines 403~407). For the preparation of the RuIn/AC catalyst, the first step is to clean the commercial activated carbon. Then, 1.0 g of clean AC was added to the mixed solution (10 mL deionized water and 5 mL ethanol) containing 20.6 mg Ru(acac)₃ and 5.6 mg In(NO₃)₃·5H₂O. The mixture was sonicated for 30 min, and then heated at 90 °C under stirring to evaporate water. The resulting mixture was calcinated at 900 °C for 3 h at a heating rate of 5 °C/min under Ar atmosphere to obtain the RuIn/AC catalysts. TEM image shows that the Ru and In in RuIn/AC catalyst are existed as metal clusters and nanoparticles, rather than atomic pairs. EDS mappings confirm that Ru and In are uniformly distributed on the carbon support (Figure R9).

Although the RuIn/AC catalyst has no N dopant, several properties of activated carbon itself, such as specific surface area, porosity, and degree of defects, can also promote catalytic performances.^[R1~R3] Therefore, we believe that the RuIn/AC catalyst can achieve excellent catalytic activity for the following reasons. (i) Obviously, the specific surface area of the activated carbon (~1500 m²/g) is larger than that of our Ru-N-In/NC material (690 m²/g). The larger the specific surface area of catalytic materials, more defects present on the substrate, the larger the contact area between reactants and catalysts, which help to distribute the active species and improve the catalytic activity. (ii) It cannot be ignored that regardless of the form (Ru nanoparticles, Ru clusters, and Ru single atoms) of Ru metal, it exhibits certain activity in the acetylene hydrochlorination reaction. Therefore, Ru-based catalysts can be emerged as the promising candidates for Hg-based catalysts. The introduction of In element further enhances the activity and stability of Ru-based catalysts.

Figure R9. (a~b) TEM images of the RuIn/AC catalyst and the corresponding EDS mappings for (c) In and (d) Ru, respectively.

However, though RuIn/AC can achieve a relatively good catalytic performances and low coke deposition mentioned by the reviewer and our experiments, its long-term stability is still far from that of the Ru-N-In/NC catalyst (more than 150 hours) (**Figure 3c** and **Figure R10**). We consider that this may be due to the lack of strong N sites for anchoring Ru and In atoms in the RuIn/AC catalyst. In contrast, for the Ru-N-In/NC catalyst, the introduction of N can favorably induce an atomic-pair structure and electrons redistribution, thereby producing electron-donating or electron-accepting sites that affect the electron distribution state of carbon materials. In addition, Ru-N-In/NC catalyst also has a higher single atom dispersion, allowing each metal atom to serve as a catalytic active site. This is also difficult to achieve for the cluster and nanoparticle catalysts. Currently, how to select the optimal

carrier to exert the intrinsic activity of Ru-based catalysts is still a subject of contention, and this topic of research still needs to be carefully investigated in the future.^[R4~R8]

Figure R10. Long-term catalytic performances of the Ru-N-In/NC and RuIn/AC catalysts.

References

- [R1] Hu, Y. et al. Mechanochemical preparation of single atom catalysts for versatile catalytic applications: A perspective review. *Mater. Today* **63**, 288-312 (2023).
- [R2] Song, W. et al. Review of Carbon Support Coordination Environments for Single Metal Atom Electrocatalysts (SACS). *Adv. Mater.* **36**, 2301477 (2024).
- [R3] Dong, C. et al. Supported Metal Clusters: Fabrication and Application in

Heterogeneous Catalysis. *ACS Catal.* **10**, 11011-11045 (2020).

[R4] Lin, R. et al. Descriptors for high-performance nitrogen-doped carbon catalysts in acetylene hydrochlorination. *ACS Catal.* **8**, 1114-1121 (2018).

[R5] Kaiser, S. K. et al. Design of Carbon Supports for Metal-Catalyzed Acetylene Hydrochlorination. *Nat. Commun.* **12**, 4016 (2021).

[R6] Chen Z. et al. Advances in Single-Atom-Catalyzed Acetylene Hydrochlorination. . *ACS Catal.* **14**, 965-980 (2024).

[R7] Martín, A. J. et al. Unifying Views on Catalyst Deactivation. *Nat. Catal.* **5**, 854–866 (2022).

[R8] Hu, Y. et al. Mechanochemical Preparation of Single Atom Catalysts for Versatile Catalytic Applications: A Perspective Review. *Mater. Today* **63**, 288-312 (2023).

10. Several studies on Ru SACs have reported the structure of Ru-N₄, but there are significant differences in catalytic activity and stability. What do the authors believe is the main reason for this?

Response: Thanks for your constructive comment. For the significant differences in catalytic activity and stability over the reported Ru-N₄ structure, we have summarized the following reasons based on the current understandings of our research team (**Figure R11**).

The introduction of N poses significant merits in regulating carbon materials. (i) The doping of N not only can regulate the physicochemical properties of carbon materials, but also change the electron affinity of the adjacent carbon atoms and electronic distribution. (ii) The introduction of N can induce a redistribution of *p*-electrons between N and C due to the higher electronegativity of N (3.04) in comparison with C (2.55), thereby producing electron-donating or electron-accepting sites that affects the electron distribution state of carbon materials. (iii) The introduction of N can also re-configure the micro-structures (e.g., binding modes of the nitrogen element on the base unit) and macro-structures (e.g., porosity, defects, surface functional groups, and electrical conductivity) of carbon catalysts, thereby changing the catalytic activity. According to the above discussions, the significant

differences in catalytic activity and stability of such similar N-regulated Ru-N₄ materials is fundamentally determined by i) the type and content of the doped N, and ii) the material structures such as defect degree, pore property, and surface chemistry.^[R1] The detailed explanations are as follows:

Figure R11. Relationship among critical factors for catalyst preparation, structure property, and catalytic performance. Taking the Ru-N₄ structure as an example.

Although Ru-N₄ configuration can be easily obtained based on the reported synthesis methods, different N source and pyrolysis temperatures achieve varied N species (such as pyridinic, pyrrolic, graphitic, and oxidized nitrogen) coordinated Ru atoms. For instance, the increase of temperature favors the generation of graphitic-N due to the transition of carbonization, which inevitably results in the loss of N content. Li et al. and Kaiser et al. systemically investigated the relationship of N content with catalytic activity, which demonstrated a significant positive correlation. The authors also showed that the increase of ZIF-8 (the common precursor for Ru-N₄ moiety) loading led to the increase of N content and an improvement of activity. However, Dai et al. revealed that the catalytic activity of NC did not increase linearly with the N content. Further, with the increase of carbonization temperature, N content significantly decreased. Recently, some publications have reported

that the pyrrolic-N constructed Ru-N₄ structures present the superior performances, as pyrrolic-N is responsible for the favorable adsorption of reactants.^[R2~R5] Thus, a specific catalyst needs to explore its appropriate carbonization temperature to achieve the best balance of nitrogen content and nitrogen type for the optimal catalytic activity.

Defect sites in Ru-N_x ($x = 2, 3, \text{ and } 4$) catalysts can facilitate HCl adsorption, thus the catalytic performances can be enhanced with the increase of defect degree, but the stability might be compromised. During the preparation of Ru-N_x catalysts, different carbonization temperature could result in distinct defective catalysts. With the increase of temperature, the defect of direct-carbonization catalysts decreased, leading to a drastically reduced catalytic activity. For the Ru-N_x catalysts, a series of characterization studies (i.e., surface graphitization, surface N/O species distribution texture, functional groups, and adsorption property) demonstrated that the catalytic activity was attributed to the binding of surface N element with defect sites. Also, the coordination between surface N elements and defect sites is also crucial for the synthesis of Ru single atom catalysts.^[R6~R9]

In addition, the structure property of Ru-N₄ catalysts (i.e., pore size, surface chemistry, and electrical conductivity) also plays a vital role in catalytic activity and stability by affecting gas diffusion, adsorption and electron transfer. For example, the pore structure of the support was proved to be another critical factor affecting the performance of acetylene hydrochlorination. During the reaction, micropores are easily blocked while mesopores and macropores were structurally stable, extending the lifetime of hierarchical NC by around fifty times compared to the blank analogs. Meanwhile, the enrichment of mesopores could offer channels for the transport and diffusion of reactants and products.^[R10~R13]

In summary, the correlations among critical preparation factors of Ru-N₄ catalysts (e.g., carbonization temperature), structure property (i.e., defect and porosity), and catalytic performance (i.e., activity and stability) are discussed to establish performance descriptors for guiding the design of efficient N-regulated catalysts. The carbonation temperature is a decisive factor in the preparation of support/catalysts. High temperature is conducive to the formation of ordered and abundant pore structure and defect sites, but it also leads to the

loss of N element, which in turn reduces the activity and stability of catalysts. Hence, an ideal temperature threshold needs to be explored for specific N-regulated materials with moderate nitrogen content, rich defects, and abundant pores. However, the low stability of such carbon materials remains a concern for current research. This is the direction that our research team has been working on for a long time. Moreover, it is still a great challenge to identify/confirm the exactly active centers in catalysts (e.g., pyrrolic-N, quaternary-N, pyridinic-N, graphitic-N, or N defects), in order to establish the precise relationship between structure and catalytic activity, which are of fundamental importance for catalytic research. These issues should be carefully investigated from at the atomic level, and need to be addressed in the future studies so as to guide the design of efficient metal catalysts toward practical acetylene hydrochlorination.^[R14~R15]

References

- [R1] Lu, F. et al. Macroporous carbon material with high nitrogen content for excellent catalytic performance of acetylene hydrochlorination. *ChemistrySelect* **5**, 878-885 (2020).
- [R2] Zhang, Q. et al. Phthalimide ligand coordination as a critical “key” for constructing chlorine-platinum-nitrogen single-site catalysts for effective acetylene hydrochlorination. *ACS Sustainable Chem. Eng.* **11**, 3103-3113 (2023).
- [R3] Li, X. et al. A novel, non-metallic graphitic carbon nitride catalyst for acetylene hydrochlorination. *J. Catal.* **311**, 288-294 (2014).
- [R4] Zhang, C. et al. Nitrogen-doped active carbon as a metal-free catalyst for acetylene hydrochlorination. *RSC Adv.* **5**, 7461-7468 (2015).
- [R5] Lin, R. et al. Descriptors for high-performance nitrogen-doped carbon catalysts in acetylene hydrochlorination. *ACS Catal.* **8**, 1114-1121 (2018).
- [R6] Lu, F. et al. High nitrogen carbon material with rich defects as a highly efficient metal-free catalyst for excellent catalytic performance of acetylene hydrochlorination. *Chinese J. Chem. Eng.* **29**, 196-203 (2021).
- [R7] Kaiser, S. K. et al. Single-Atom Catalysts Across the Periodic Table. *Chem. Rev.* **120**, 11703-11809 (2020).

- [R8] Chen, Z. et al. Advances in Single-Atom-Catalyzed Acetylene Hydrochlorination. *ACS Catal.* **14**, 965-980 (2024).
- [R9] Wang, J. et al. Effect of carbon defects on the nitrogen-doped carbon catalytic performance for acetylene hydrochlorination. *Appl. Catal. A-Gen.* **564**, 72-78 (2018).
- [R10] Lan, G. et al. Defective graphene@diamond hybrid nanocarbon material as an effective and stable metal-free catalyst for acetylene hydrochlorination. *Chem. Commun.* **55**, 1430-1433 (2019).
- [R11] Qiu, Y. et al. Defect-rich activated carbons as active and stable metal-free catalyst for acetylene hydrochlorination. *Carbon* **146**, 406-412 (2019).
- [R12] Zhao, F. and Kang, L. The neglected significant role for graphene-based acetylene hydrochlorination catalysts-intrinsic graphene defects. *ChemistrySelect* **2**, 6016-6022 (2017).
- [R13] Kaiser, S. K. et al. Nitrogen-doped carbons with hierarchical porosity via chemical blowing towards long-lived metal-free catalysts for acetylene hydrochlorination. *ChemCatChem* **12**, 1922-1925 (2020).
- [R14] Kaiser, S. K. et al. Design of Carbon Supports for Metal-Catalyzed Acetylene Hydrochlorination. *Nat. Commun.* **12**, 4016 (2021).
- [R15] Wang, X. et al. Progress of *p*-Block Element-Regulated Catalysts for Acetylene Hydrochlorination. *Coordin. Chem. Rev.* **500**, 215541 (2024).

Reviewer 3

In this manuscript, the authors reported the acetylene hydrochlorination by asymmetric Ru-In atomic pairs with excellent yields and stability higher than 600 h. A deep study of the catalyst by experimental and theoretical techniques was performed to obtain that a *d-p* orbital interaction between Ru and In is the responsible of the selective absorption of acetylene in In and HCl in Ru, weaker than with pure Ru. This catalyst works better than the corresponding RuN₄/NC, that aggregates due to the over-chlorination. However, in order to complete the study, is necessary a revision of the manuscript.

Response: We would like to sincerely thank you for your careful review and outstanding contributions for our manuscript. We have addressed issues one by one in response to your comments and supplemented characterization results and analyses according to your comments. The quality of the revised manuscript has been improved with the help of editor and reviewers. We sincerely appreciate your valuable comments.

Catalyst characterization

1. It surprises me how the authors named the catalysts. For the monometallic ones (Ru and In) they denoted as RuN₄/NC and InN₄/NC, remarking that the metals are coordinated to 4 N from the support. However, the dimer is noted as RuIn/NC, avoiding that Ru and In are coordinated through a N since there is no evidence of Ru-In bond. If the active specie is Ru-N-In, why not name it with the N?

Response: Thanks for your valuable suggestion. We have checked the manuscript and supplementary information thoroughly, and then revised the “RuIn/NC” to “Ru-N-In/NC”.

2. In Supplementary figure 54 the authors have a picture that reveals the agglomeration of Ru in RuN₄/NC catalyst after reaction, but they do not show the same image for RuIn/NC catalyst, that is the important catalyst, so they can not demonstrate that they do not observe agglomeration of the active species after reaction.

Response: Thank you for your careful comment. We have added the AC-HAADF-STEM image of the Ru-N-In/NC catalyst after 3 hours in **Supplementary Figure 57** and 600 h in **Figure 3c**, both of the two images confirm that the Ru-N-In pairs are stable during the acetylene hydrochlorination reaction.

The revised description: AC-HAADF-STEM analysis of RuN₄/NC-3h indicated that the Ru centers aggregated from single atoms to nanoparticles with sizes up to ~4 nm, whereas the single-atom pairs were still retained on Ru-N-In/NC-3 h catalyst (Supplementary Fig. 57), and even on Ru-N-In/NC-600 h catalyst (**Figure 3c**). (Lines 292~295)

Supplementary Figure 57. AC-HAADF-STEM image of RuN₄/NC-3h and Ru-N-In/NC-3h.

Figure 3c. AC-HAADF-STEM image of Ru-N-In/NC-600 h.

Catalytic study

3. The authors achieve a deep characterization of the catalyst but regarding the catalytic study it seems that they only present the results without an analysis of them. For example, 180 °C reaction temperature is better than the higher ones (210 and 240 °C) What happens at these temperatures? Is it a problem of deabsorption of HCl or acetylene in the catalyst?

Response: Thanks for your critical comment. The process of acetylene hydrochlorination is relatively complex, mainly involving the following reaction processes of R1~R6.

Main reaction:

Non-polymerization side reactions:

Polymerization side reactions:

We calculated the Gibbs free energy of each reaction based on the following equation:

$$\Delta_r G_m^0 (298 \text{ K}) = \Delta_r H_m^0 (298 \text{ K}) - T\Delta_r S_m^0 (298 \text{ K}) \quad (\text{R7})$$

Table R2. Calculation results of thermodynamic parameters for each reaction.

Reaction	$\Delta_r S_m^0/\text{J/mol}\cdot\text{K}$	$\Delta_r H_m^0/\text{kJ/mol}$	$\Delta_r G_m^0/\text{kJ/mol}$
R1	-123.9	-98.9	-61.9
R2	-145.8	-71.0	-27.5
R3	-144.9	-71.1	-27.9
R4	-146.2	-155.3	-111.8
R5	-122.5	-148.9	-112.4
R6	-147.7	-184.3	-140.3

From a thermodynamic point of view, since the acetylene hydrochlorination reaction is exothermic, too high or too low a temperature could lead to a decrease in acetylene conversion (**Table R2**). To ensure that the acetylene conversion is not less than 99%, the reaction temperature should be less than 200 °C according to actual industrial production. In addition, it has been reported that once the reaction temperature exceeds 200 °C, it is very easy to induce the polymerisation side reaction (especially Reaction 6) leading to the polymerisation of vinyl chloride monomer with unreacted acetylene.^[R1~R3]

We also checked the adsorption of acetylene over the Ru-N-In/NC catalyst through *in-situ* DRIFTS technique, as shown in **Figure R12**. It can be seen that the strength of acetylene adsorption generally decreases with increasing reaction temperature (from 210 to 240 °C) in comparison with the strength of acetylene adsorption at 180 °C (**Figure 4b**), indicating that the adsorption of acetylene was partially inhibited.

In addition, we further conducted the HCl-TPD curves to confirm the absorption of HCl at the temperature of 210 and 240 °C. As indicated in **Figure R13**, the desorption temperature of HCl gradually decreased from 173 to 161 °C as the reaction temperature increased from 180 to 240 °C. Furthermore, a remarkable decline of the desorption peak area for Ru-N-In/NC at 210 °C (3823) and 240 °C (3184) was observed compared to the

case at 180 °C (4467.3), indicating that the increase in reaction temperature weakens the adsorption of HCl on the Ru-N-In/NC catalysts. Overall, the weakened adsorption for acetylene and HCl at the higher temperature of 210 and 240 °C can result in the lower conversion efficiency for acetylene hydrochlorination.^[R4-R5]

Figure R12. In situ DRIFTS of acetylene adsorption as a function of time at 210 and 240 °C over Ru-N-In/NC.

Figure R13. HCl-TPD curves of the Ru-N-In/NC catalyst at the temperature of 180, 210, and 240 °C.

References

- [R1] Wang, L. et al. Thermodynamic analysis of acetylene hydrochlorination reaction system. *Chem. Ind. Eng. Prog.* **3**, 1-4 (2014).
- [R2] Kaiser, S. et al. Performance Descriptors of Nanostructured Metal Catalysts for Acetylene Hydrochlorination. *Nat. Nanotechnol.* **17**, 606-612 (2022).
- [R3] Zhang, T. et al. Excess Copper Chloride Induces Active Sites over Cu-Ligand Catalysts for Acetylene Hydrochlorination. *ACS. Catal.* **13**, 8307-8316 (2023).
- [R4] Lu, F. et al. High-Density Pyridine-FeN₄ Active Sites for Acetylene Hydrochlorination. *J. Catal.* **422**, 69-76 (2023).
- [R5] Zhang, M. et al. Construction of Ru-N Single Sites for Effective Acetylene Hydrochlorination: Effect of Polyethyleneimine Modifiers. *ACS Sustainable Chem. Eng.* **10**, 13991-14000 (2022).

4. If the active specie is the asymmetric Ru-In atomic pairs, theoretically, the best ratio of starting materials should be Ru: In = 1:1 in order to obtain a higher amount of dimers; however, the authors claimed that the best ratio Ru/In is 5, confirmed by ICP. There is an excess of Ru in the catalyst. Did you observe Ru single atoms in the sample or clusters? Since you claim in line 128 that a fraction of Ru and In appear in intimate atomic pairs, do you know the amount of Ru and In that are forming the dimers?

Response: Thanks for your rigorous comment. As for the optimal Ru/In weight ratio of ~ 5:1, we believe that more Ru-N-In atom pairs can be generated in this case, since there are indeed some Ru single atom sites (marked by yellow circles) can be identified in the AC-HAADF-STEM image of Ru-N-In/NC catalyst (**Figure R14**)

Figure R14. AC-HAADF-STEM image of Ru-N-In/NC, in which the signal-atom Ru-In pairs and Ru single atoms are highlighted by yellow ovals/boxes and circles, respectively.

To further prove the above point of the generation of more Ru-N-In atom pairs when the Ru/In weight ratio is ~ 5:1, the DRIFTS for CO adsorption studies were conducted to

quantify different Ru active centers (Ru-N₄ and Ru-N-In), as displayed in **Figure R5**. The band at 2129 and 2121 cm⁻¹ could be attributed to multicarbonyl species adsorbed on partially oxidized Ru^{δ+} sites (donated as Ru^{δ+}(CO)_x with *x* = 2 or 3). Compared with the Ru-N₄ sample, the peak at 2121 cm⁻¹ can be attributed to the interaction between CO and Ru-N-In sites, while the peak at 2129 cm⁻¹ is assigned to the interaction between Ru-N₄ and CO. The broad band locates in the range of 2000~1920 cm⁻¹, and is centered at around 1971 cm⁻¹, which could be ascribed to the linear-adsorbed CO on the boundary between Ru sites and the support or bridge-adsorbed CO on Ru sites.^[R1~R5]

Figure R5. In situ DRIFTS spectra and the peak deconvolution analysis of CO adsorption on the Ru-N-In/NC and RuN₄/NC catalysts at 180 °C.

Subsequently, we performed peak deconvolution analysis for the DRIFTS spectra to further quantify different active centers (Ru-N₄ and Ru-N-In). As shown in **Figure R6** and **Table R1**, a volcano-type correlation was observed between the Ru-N-In content and Ru/In ratio, confirming the highest level of Ru-N-In content (79.6%) was achieved in the case of Ru/In ratio of ~5:1. Furthermore, the initial acetylene conversion efficiency and Ru-N-In

content follows a linear correlation ($R^2 = 0.94$), where the highest catalytic activity was obtained at the highest levels of Ru-N-In species. Only ~50% of the Ru atoms in the Ru-N-In/NC catalysts synthesized by our methods are in the Ru-N₄ form at the Ru/In ratio of 1. In this case, too much Ru-N₄ species anchored on the surface of Ru-N-In/NC is not beneficial for enhancing the activity and stability due to the inevitable influence of over-chlorination. Moreover, the chlorinated Ru-N₄ also triggers the migration and loss of Ru atoms. In contrast, the more Ru-N-In species can perfectly address this issue to sustain a superior performance.

In conclusion, based on current experimental and characterization results, the optimal Ru/In ratio was identified as ~ 5:1 to obtain more Ru-N-In atom pairs. In future studies, our research team will further involve in increasing the content of atom pairs to achieve the better performance in acetylene hydrochlorination.

Figure R6. Ru-N-In content as a function of the Ru/In ratio, and the initial acetylene conversion efficiency as a function of the Ru-N-In species content.

Table R1. Fitting parameters derived from In situ DRIFTS spectra of CO adsorption of the different Ru-N-In/NC and Ru/NC catalysts.

Catalysts	Ru-N-In		Ru-N ₄	
	Position (cm ⁻¹)	Content (%)	Position (cm ⁻¹)	Content (%)
Ru-N-In/NC Ru : In = 10 : 1		63.3		36.7
Ru-N-In/NC Ru : In = 5 : 1		79.6		20.4
Ru-N-In/NC Ru : In = 2 : 1	2121	58.9	2129	41.1
Ru-N-In/NC Ru : In = 1 : 1		53.8		46.2
RuN ₄ /NC	/	/		100

References

- [R1] Lyu, S. et al. Dopamine Sacrificial Coating Strategy Driving Formation of Highly Active Surface-Exposed Ru Sites on Ru/TiO₂ Catalysts in Fischer-Tropsch Synthesis. *Appl. Catal. B-Environ.* **278**, 119261 (2020).
- [R2] Abdel-Mageed, A. M. et al. Selective CO Methanation on Ru/TiO₂ catalysts: Role and Influence of Metal-Support Interactions. *ACS Catal.* **5**, 6753-6763 (2015)
- [R3] Panagiotopoulou, P. et al. Mechanistic Study of the Selective Methanation of CO over Ru/TiO₂ Catalyst: Identification of Active Surface Species and Reaction Pathways. *J. Phys. Chem. C* **115**, 1220-1230 (2011).
- [R4] Elmasides, C. et al. Partial Oxidation of Methane to Synthesis Gas over Ru/TiO₂ Catalysts: Effects of Modification of the Support on Oxidation State and Catalytic Performance. *J. Catal.* **198**, 195-207 (2001).
- [R5] Davydov, A. A. and Bell, A. T. An Infrared Study of NO and CO Adsorption on a Silica-Supported Ru Catalyst. *J. Catal.* **49**, 332-344 (1977).

5. In Figure 35 the authors show the activity of other bimetallic catalyst obtained using the same procedure. Do the authors have any characterization data to confirm that they also obtain the asymmetric M-In atomic pairs?

Response: Thanks for your careful comment. To validate the asymmetric M-In atomic pairs on the Au/Pt/Pd-In bimetallic catalysts, we firstly performed AC-HAADF-STEM analysis of the AuIn/NC, PtIn/NC, and PdIn/NC catalysts. AC-HAADF-STEM images exhibited that the Au/Pt/Pd and In atoms are well-dispersed as atomic pairs throughout the whole carbon matrix, where the white bright dimeric spots (marked by yellow circles) were identified as Au-In, Pt-In, and Pd-In atomic pairs (**Figure R15**).

Figure R15. AC-HAADF-STEM images of (a) AuIn/NC, (c) PtIn/NC, and (c) PdIn/NC samples, in which the Au/Pt/Pd-In atomic pairs are marked by the yellow circles.

Subsequently, we conducted the X-ray absorption spectroscopy to further confirm the coordination environments and local structures of the Au-In/NC, Pt-In/NC, and Pd-In/NC catalysts (**Figures R16~R18** and **Tables R3~R5**). XANES schemes indicated that Au, Pt, Pd, and In are atomically dispersed on the NC supports, and in particular, Au-In, Pt-In, and Pd-In appear in intimate atom pairs. Furthermore, EXAFS and the quantitative structural parameters analysis demonstrated that each Au/Pt/Pd atom coordinates with three N atoms, and each In atom is coordinated with four N atoms. Thus, according to the experimental results, it can be concluded that the synthetic strategies and methods for Ru-N-In/NC can be applied to other noble metal (Au, Pt, and Pd) catalysts.

Figure R16. Au and In K-edge XAFS analysis of the Au-In/NC catalyst, including normalized XANES, and Fourier-transform XAFS spectra.

Table R3. EXAFS fitting parameters at the Au and In *K*-edge for AuIn/NC sample.

Samples	Coordination	CN ^a	R(Å) ^b	σ ² (Å ²) ^c	ΔE ₀ (eV) ^d	R factor
Au foil	Au-Au	12*	2.859±0.004	0.0076±0.0012	3.5±0.2	0.0013
AuIn/NC	Au-N	3.1±0.3	2.098±0.002	0.0201±0.0030	2.0±1.3	0.0015
In foil	In-In	12*	3.239±0.009	0.0320±0.0014	-1.2±0.5	0.0036
AuIn/NC	In-N	4.2±0.3	2.133±0.001	0.0071±0.0012	2.2±0.4	0.0043

Figure R17. Pt/In K-edge XAFS analysis of the Pt-In/NC catalyst, including normalized XANES, and Fourier-transform XAFS spectra.

Table R4. EXAFS fitting parameters at the Pt and In *K*-edge for PtIn/NC sample.

Sample	Coordination	CN ^a	R(Å) ^b	$\sigma^2(\text{\AA}^2)^c$	$\Delta E_0(\text{eV})^d$	R factor
Pt foil	Pt-Pt	12*	2.768±0.005	0.0048±0.0002	4.2±2.1	0.0005
PtIn/NC	Pt-N	2.9±0.5	1.978±0.004	0.0035±0.0015	4.8±1.3	0.0058
In foil	In-In	12*	3.230±0.0012	0.0129±0.0024	1.6±0.4	0.0015
PtIn/NC	In-N	3.7±0.2	2.132±0.002	0.0042±0.0023	2.3±1.2	0.0117

Figure R18. Pd/In K-edge XAFS analysis of the Pd-In/NC catalyst, including normalized XANES, and Fourier-transform XAFS spectra.

Table R5. EXAFS fitting parameters at the Pd and In *K*-edge for PdIn/NC sample.

Sample	Coordination	CN^a	$R(\text{\AA})^b$	$\sigma^2(\text{\AA}^2)^c$	$\Delta E_0(\text{eV})^d$	R factor
Pd foil	Pd-Pd	12*	2.742±0.002	0.0048±0.0002	-2.2±0.8	0.0008
PdIn/NC	Pd-N	3.3±0.3	2.098±0.005	0.0041±0.0012	-5.2±0.8	0.0013
In foil	In-In	12*	3.241±0.025	0.0325±0.0017	1.9±0.3	0.0074
PdIn/NC	In-N	4.3±0.4	2.129±0.006	0.0047±0.0011	2.2±2.1	0.0058

Note that the explanations of several abbreviations shown in **Tables R3~R5** are as follows: ^a*CN*, coordination number; ^b*R*, the distance to the neighboring atom; ^c σ^2 , the mean square relative displacement; ^d ΔE_0 , inner potential correction; *R* factor indicates the goodness of the fit. S_0^2 was fixed to 0.863 and 0.788, according to the experimental EXAFS fit of Au/Pt/Pd foil and In foil by fixing *CN* as the known crystallographic value. * This value was fixed during the EXAFS fitting, based on the known structure of Au, Pt, Pd, and In. A reasonable range of EXAFS fitting parameters: $0.700 < S_0^2 < 1.000$; $CN > 0$; $\sigma^2 > 0 \text{ \AA}^2$; $|\Delta E_0| < 10 \text{ eV}$; *R* factor < 0.02 .^[R1~R3]

References

- [R1] Ravel, B. and Newville, M. ATHENA, ARTEMIS, HEPHAESTUS: data analysis for X-ray absorption spectroscopy using IFEFFIT. *J. Synchrotron Rad.* **12**, 537-541 (2005).
- [R2] Funke, H. et al. Wavelet Analysis of Extended X-ray Absorption Fine Structure Data. *Phys. Rev. B* **71**, 094110 (2005).
- [R3] Zabinsky, S. I. et al. Multiple-Scattering Calculations of X-Ray-Absorption Spectra. *Phys. Rev. B* **52**, 2995-3009 (1995).

6. The catalytic test was analyzed by gas chromatography, but the authors forgot to describe the column used, necessary to reproduce the procedure.

Response: Thanks for your careful comment. We have described the column used, and then revised the corresponding description. **(Lines 444~446)**

The revised version: The effluent gas was passed into NaOH solution and a dryer, followed by analyzing with a gas chromatograph (GC, PANNA A60) equipped with a flame ionization detector (FID) and a **KB-624 column (30 m × 0.32 mm × 1.8 μm)**.

7. The other products obtained with InN₄/NC are 1,1-dichloroethane and 1,2-dichloroethane. How they can confirm that these are the products obtained? No description facilitated in the manuscript or supporting.

Response: Thank you for your careful check and accurate guidance. Typically, acetylene hydrochlorination reactions require a slight excess of HCl into the reactor to activate the active site. Massive publications and actual industrial production have pointed out that the optimal ratio of HCl/C₂H₂ is 1.05~1.2 : 1. Therefore, when there is an excess of HCl in the reaction system, the oxidation of HCl or the further hydrochlorination of vinyl chloride can result in forming the by-product of dichloroethane (such as 1,1-dichloroethane (R1) and 1,2-dichloroethane (R2)).^[R1~R5] We have added the product analysis results obtained from gas chromatography, and a small peak of dichloroethane can be clearly observed at 2.142 min. We have added the result in Supporting Information of **Supplementary Figure 34**.

Supplementary Figure 34. The product analysis through gas chromatography over the InN₄/NC catalyst.

The revised version: The mainly by-product, which may originate from the oxidation of HCl or further hydrochlorination of vinyl chloride, was identified as dichloroethane (1,1-dichloroethane and 1,2-dichloroethane) through gas chromatograph. (Lines 161~165)

References

[R1] Peng, J. et al. Manipulating Micro-Electric Field and Coordination-Saturated Site Configuration Boosted Activity and Safety of Frustrated Single-Atom Cu/O Lewis Pair for Acetylene Hydrochlorination. *Nano Res.* **16**, 6178-6186 (2023).

[R2] Fan, Y. et al. Metal-Organic Frameworks Encaged Ru Single Atoms for Rapid Acetylene Harvest and Activation in Hydrochlorination. *ACS Appl. Mater. Interfaces* **15**, 24701-24712 (2023).

[R3] Shang, S. et al, Highly Efficient Ru@IL/AC To Substitute Mercuric Catalyst for Acetylene Hydrochlorination. *ACS Catal.* **7**, 3510-3520 (2017).

[R4] Kaiser, S. K. et al. Sustainable Synthesis of Bimetallic Single Atom Gold-Based Catalysts with Enhanced Durability in Acetylene Hydrochlorination. *Small* **17**, 2004599 (2021).

[R5] Martín, A. J. et al. Unifying Views on Catalyst Deactivation. *Nat. Catal.* **5**, 854-866 (2022).

References

8. In the Introduction, page 2 line 24, the authors missed some references regarding the hydrochlorination of acetylene with Au, one of the most used catalyst to substitute Hg: *J. Catal.* 1985, 96, 292; *Appl. Catal.* 1985, 17, 155; *J. Catal.* 2007, 250, 231; *Angew.*, 2017, 56, 6435.

Response: Thanks for your comment. We have added these important references in our

reference list of 2#, 4#, 5#, and 8#.

The revised version:

[2] Hutchings, G. J. and Grady, D. T. Hydrochlorination of Acetylene: The effect of Mercuric Chloride Concentration on Catalyst Life. *Appl. Catal.* **17**, 155-160 (1985).

[4] Oliver-Meseguer, J. et al. Partial Reduction and Selective Transfer of Hydrogen Chloride on Catalytic Gold Nanoparticles. *Angew. Chem. Int. Ed.* **56**, 6435-6439 (2017)

[5] Hutchings, G. J. Vapor Phase Hydrochlorination of Acetylene: Correlation of Catalytic Activity of Supported Metal Chloride Catalysts. *J. Catal.* **96**, 292-295 (1985)

[8] Conte, M. et al. Hydrochlorination of Acetylene Using a Supported Gold Catalyst: A Study of the Reaction Mechanism. *J. Catal.* **250**, 231-239 (2007)

Other issues

9. The graphs should be represented also with error bars.

Response: Thanks for your careful comment. We have added error bars in the following revised figures.

Figure 3a. C₂H₂ conversions of Ru-N-In/NC, RuN₄/NC, InN₄/NC and NC.

Supplementary Figure 28. Acetylene conversion efficiency and VCM selectivity of Ru-N-In/NC at different temperature (120~240 °C).

Supplementary Figure 29. Acetylene conversion efficiency and VCM selectivity of Ru-N-In/NC at different Ru/In ratios (1.0~10).

Supplementary Figure 30. Acetylene conversion efficiency and VCM selectivity of Ru-N-In/NC at different $GHSV(C_2H_2)$ of 50~1000 h⁻¹.

Supplementary Figure 33. VCM selectivity of Ru-N-In/NC, RuN₄/NC, and InN₄/NC.

Supplementary Figure 39. Acetylene conversion efficiency and VCM selectivity for RuIn/AC.

Supplementary Figure 40. Acetylene conversion efficiency and VCM selectivity for Ru-N-In/NC and HgCl₂/AC, where HgCl₂ loading was 10 wt.% and 5 wt.%.

10. Line 24: In recent, (in recent times, in recent years...?)

Response: Thanks for your careful comment. We have checked the whole manuscript and revised the “In recent” to “In recent years”. (Lines 24~26)

The revised version: In recent years, ruthenium single-atom catalysts (Ru SACs) have emerged as the promising candidates due to their excellent chlorine affinity and flexible control of active-site architectures.

11. Line 65: polyaniline (PAN). Lines 365 and 368: PANI.

Response: Thanks for your careful comment. We have checked and revised the “PANI” to “PAN”. (Lines 374~378)

The revised version: To form higher molecular weight PAN, the polymer slurry was continuously stirred for 24 h at room temperature to complete polymerization process, subsequently filtered, and washed with deionized water and ethanol. The formed PAN was dried in static air at 120 °C for 12 h and finally carbonized at 800 °C (5 °C/min) in N₂

atmosphere for 1 h to yield NC.

12. Scheme 1 and Figure 1: RuN₄/NC, InN₄/NC (subscript).

Response: Thanks for your careful comment. We have revised **Figure 1c** and **Scheme 1**.

The revised figures:

Figure 1c. Powder XRD patterns of Ru-N-In/NC, RuN₄/NC, InN₄/NC, and NC.

(a) Over-chlorination process

(b) This study: Asymmetric catalysis

Scheme 1. (a) The over-chlorination process of the symmetrical RuN₄/NC moiety during

the hydrochlorination reaction, and (b) the corresponding solution through breaking the geometric symmetry to construct a unique asymmetric Ru-N-In/NC configuration.

13. In supporting Figure 25, the scheme shows a bottle of H₂S not used in the experiment, or at least not mentioned in the manuscript. Please, delete to avoid misunderstandings.

Response: Thanks for your careful comment. We have revised the H₂S bottle to CO₂ bottle.

Supplementary Figure 25. Schematic of experimental apparatus for acetylene hydrochlorination.

14. Supporting Figure 35: The obtained data of the there (??) samples are the same as the reaction conditions of RuIn/NC.

Response: Thanks for your careful comment. We have corrected “there samples” to “three samples”.

The revised version: Supplementary Figure 38. Comparison of TOF ($\text{mol}_{\text{C}_2\text{H}_2}/\text{mol}_{\text{metal}}/\text{h}$) of Ru-N-In/NC with the AuIn/NC, PdIn/NC, and PtIn/NC catalysts. Note that the synthesis

of AuIn/NC, PdIn/NC, and PtIn/NC follows the similar procedure as that of Ru-N-In/NC, shown in **Supplementary Figure 1**. The obtained data of the **three samples** are the same as the reaction conditions of Ru-N-In/NC.

15. Supporting Figure 46: please, indicate which colour is Ru and In.

Response: Thanks for your careful comment. We have checked the entire Manuscript and Supplementary Information, and then added the corresponding color descriptions.

The revised version: White, gray, blue, orange, and pink balls represent H, C, N, Ru, and In atoms, respectively.

16. Reference 29: paper volume in bold.

Response: Thanks for your careful comment. We have revised the Reference 29.

The revised version: [29] Yan, L. et al. Atomically Precise Electrocatalysts for Oxygen Reduction Reaction. *Chem* **9**, 280-342 (2023).

Special thanks for all the comments and contributions!

Sincerely,

Pengfei Xie

Professor,

College of Chemical and Biological Engineering

Zhejiang University

REVIEWER COMMENTS

Reviewer #1 (Remarks to the Author):

Most of the questions were addressed. There is one more concern about Supplementary Table 4, which was refitted by considering the Ru-Ru and In-In in the second shell. The Ru-N-Ru and In-N-In are suggested to be changed to Ru-N-Ru/In and In-N-In/Ru, respectively. The atomic numbers of the In and Ru are too close to be distinguished by the EXAFS. Therefore, the formation of the Ru-N-In or In-N-Ru cannot be proved by the EXAFS data. The HAADF-STEM, CO adsorption DRIFT and DFT calculations might be helpful. Besides, the ΔE_0 of the Ru-N-Ru/In is too large to be reasonable, indicating the Ru and/or In cannot be found in the second shell. As there were excess Ru single atoms without pairing with In, the N and/or C might be dominated in the second shell. As to the In-N-In/Ru, the data were seemed reasonable since the In and/or Ru could be well fitted in the second shell. This result indicated most of the In might be paired with In and/or Ru in the second shell. One more question, the data fitting range of R ($1.0 \leq R (\text{\AA}) \leq 2.5$) should be double checked. The distances of Ru-N-Ru/In and In-N-In/Ru were somewhat out of this range.

Reviewer #2 (Remarks to the Author):

We appreciate the careful revisions by the authors. There is only one issue that is still confusing. The authors provide the analysis of DRIFTS spectra for CO adsorption in their revised manuscript, to show that the highest level of Ru-N-In content ($\sim 80\%$) could be achieved in the case of Ru/In ratio of $\sim 5:1$. However, I am quite concerned that Why can the content of Ru-N-In sites ($\sim 80\%$) even exceeds that of Ru-N₄ ($\sim 20\%$) sites when Ru is significantly excess (Ru : In = 5 : 1)? Where is the excess Ru?

Reviewer #3 (Remarks to the Author):

I want to thank the authors for their improved manuscript and the methodical response to the questions provided by the reviewers. The authors have answered all of them rigorously and without any ambiguity. Regarding the new manuscript, there is still some room for improvement, such as, for example, a better efficiency in the synthesis of Ru:In pairs, the manuscript is complete for publication.

I would like to remember to add error bars to the new graphs, especially the ones with trend lines (for example, Supplementary Figures 35, 36, 51, Figure 4, etc.).

Point by Point Response to Reviewer Comments

Reviewer 1#

Most of the questions were addressed. There is one more concern about Supplementary Table 4, which was refitted by considering the Ru-Ru and In-In in the second shell. The Ru-N-Ru and In-N-In are suggested to be changed to Ru-N-Ru/In and In-N-In/Ru, respectively. The atomic numbers of the In and Ru are too close to be distinguished by the EXAFS. Therefore, the formation of the Ru-N-In or In-N-Ru cannot be proved by the EXAFS data. The HAADF-STEM, CO adsorption DRIFT and DFT calculations might be helpful. Besides, the ΔE_0 of the Ru-N-Ru/In is too large to be reasonable, indicating the Ru and/or In cannot be found in the second shell. As there were excess Ru single atoms without paring with In, the N and/or C might be dominated in the second shell. As to the In-N-In/Ru, the data were seemed reasonable since the In and/or Ru could be well fitted in the second shell. This result indicated most of the In might be pared with In and/or Ru in the second shell. One more question, the data fitting range of R ($1.0 \leq R (\text{\AA}) \leq 2.5$) should be double checked. The distances of Ru-N-Ru/In and In-N-In/Ru were somewhat out of this range.

Response: We appreciate the reviewer for the valuable contributions to our manuscript. According to your suggestions, we have considered Ru-N-Ru/In and In-N-In/Ru as the comprehensive models and fitted the raw XAS data again. The obtained results in the **Revised Supplementary Table 4** demonstrate to have a higher accuracy compared to the previous analysis, indicating that the Ru-N-In/NC catalysts may indeed have both the structures of Ru-N-Ru/In and In-N-In/Ru within the carbon matrix. In addition, we have checked the data fitting range of R ($1.0 \leq R (\text{\AA}) \leq 2.5$) in the original manuscript, which is a typo and then corrected as $1.0 \leq R (\text{\AA}) \leq 4.0$ in the revised table, as shown in the description below **Revised Supplementary Table 4**. Accordingly, we have revised the discussions related to the EXAFS characterization by removing some arbitrary comments.

To prove the higher possibility of the presence of Ru-N-In pairs, we added additional

AC-HAADF-STEM images (**Supplementary Figures 6~7**), CO-DRIFT adsorption tests (**Figure R1**), and DFT calculations (**Figure R2**) in the **First Revised Manuscript** and **Supplementary Information**. Briefly, the total 19 atom pairs decorated on the different surfaces marked in the AC-HAADF-STEM images indicate that the Ru-N-In/NC catalysts have Ru-In/Ru atomic pairs, and the average distance between the two neighboring atomic sites was determined as ~ 0.37 nm. Besides, the results of DRIFT for CO adsorption demonstrated that the main active center over the Ru-N-In/NC catalysts is dimeric heterostructure (i.e., Ru \cdots In). Moreover, we established the structures of Ru-N-Ru, In-N-In, and Ru-N-In through density functional theory, and then compared their Gibbs free energy. The asymmetric Ru-N-In dual-atom structure has the lowest free energy of -2.793 eV, indicating that Ru-N-In dual-atom structure is the most stable configuration in comparison with the Ru-N-Ru and In-N-In. Therefore, combining the above multiple characterizations and DFT calculation, we tend to believe that the Ru-N-In structures are dominated in the Ru-N-In/NC catalyst.

Revised Supplementary Table 4 EXAFS fitting parameters at the Ru and In *K*-edge for Ru-N-In/NC.

Sample	Coordination	CN	R (Å)	σ^2 (Å ²)	ΔE_0 (eV)	R factor
Ru-N-In/NC	Ru-N	3.4±0.4	2.008±0.011	0.0043±0.0019	0.2±1.6	0.0014
	Ru-N-Ru/In	1.7±0.5	3.610±0.017	0.0102±0.0049	-8.5±0.5	
	In-N	4.1±0.7	2.130±0.006	0.0056±0.0016	0.9±0.3	0.0013
	In-N-In/Ru	1.8±0.6	3.608±0.028	0.0132±0.0059	-9.6±2.1	

Note that *CN*: coordination numbers; *R*: bond distance; σ^2 : Debye-Waller factors; ΔE_0 : the inner potential correction; *R* factor: goodness of fit. A reasonable range of EXAFS fitting parameters: $0.700 < S_0^2 < 1.000$; $CN > 0$; $\sigma^2 > 0$ Å²; $|\Delta E_0| < 10$ eV; *R* factor < 0.02 . Fitting range: $3.0 \leq k$ (/Å) ≤ 13.5 and $1.0 \leq R$ (Å) ≤ 3.0 (Ru foil); $3.0 \leq k$ (/Å) ≤ 10.5 and $1.0 \leq R$ (Å) ≤ 4.0 (**Ru-N-In/NC**); $3.0 \leq k$ (/Å) ≤ 11.0 and $2.0 \leq R$ (Å) ≤ 4.0 (In foil); $3.0 \leq k$ (/Å) ≤ 10.5 and $1.0 \leq R$ (Å) ≤ 4.0 (**Ru-N-In/NC**).

Supplementary Figure 6. AC-HAADF-STEM image of the Ru-N-In/NC sample, and the distance (nm) of isolated Ru and In dual single-atom sites from 4# to 11#.

Supplementary Figure 7. AC-HAADF-STEM image of the Ru-N-In/NC sample, and the distance (nm) of isolated Ru and In dual single-atom sites from 12# to 19#.

Figure R1. In situ DRIFTS spectra and the peak deconvolution analysis of CO adsorption on the Ru-N-In/NC and RuN₄/NC catalysts at 180 °C.

Figure R2. Structure of Ru-N-Ru, In-N-In, and Ru-N-In models and their free energies. Orange, pink, blue, and gray spheres represent Ru, In, N, and C atoms, respectively.

Reviewer 2#

We appreciate the careful revisions by the authors. There is only one issue that is still confusing. The authors provide the analysis of DRIFTS spectra for CO adsorption in their revised manuscript, to show that the highest level of Ru-N-In content (~80%) could be achieved in the case of Ru/In ratio of ~5:1. However, I am quite concerned that why can the content of Ru-N-In sites (~80%) even exceeds that of Ru-N₄ (~20%) sites when Ru is significantly excess (Ru : In = 5 : 1)? Where is the excess Ru?

Response: Thanks for your careful review and recognition of our revision work. Although the Ru-In atomic pair was determined as the main structure in the Ru-N-In/NC catalyst through the DRIFTS spectra of CO adsorption, we have to acknowledge that the results are semi-quantitative, which may be influenced by various factors such as extinction coefficient due to that this analysis is based on Lambert-Beer law. Thus, the population of Ru-N-In pairs and Ru single atoms obtained by calculating the peak areas are not exactly equivalent to the actual mol ratio. Moreover, the combined experimental and theoretical analysis also demonstrate the prevalence of Ru-N-In pairs, such as the more rigorous fitting analysis of EXAFS data (**Revised Supplementary Table 4**) demonstrates the presence of Ru-N-In structure and the DFT calculation (**Figure R3**) indicates that the asymmetric Ru-N-In dual-atom pair structure has the lower free energy of -2.793 eV in comparison with Ru-N₄ configuration (-2.712 eV).

In addition to the Ru atom coordinated to In atom, we consider that the excess Ru may exist in the following forms: (i) Ru-N_x species. Based on the reported publications and our experimental results,^[R1~R5] Ru single atoms (Ru-N_x ($x = 2, 3, \text{ and } 4$), mainly Ru-N₄ species) are unavoidably produced during the synthesis processes. What's more, there are indeed some Ru single atom sites (marked by red circles) can be identified in AC-HAADF-STEM image of the Ru-N-In/NC catalysts (**Figure R4**). (ii) Ru-C species. Several previous studies have demonstrated that the Ru-N single-atom sites or Ru-N-M (M refers to metal elements) dual-atom sites can be formed at approximately 800 °C through pyrolysis treatment.^[R6~R8] However, the gradual loss of N dopants always occurred with the

increasing of pyrolysis temperature. Therefore, by controlling the pyrolysis temperature, the loss of N dopants was controlled, and the local environment and coordination structure of Ru center were spontaneously controlled. Since our catalyst was calcinated at 900 °C under Ar atmosphere, some Ru atoms can escape the control of N dopants, and then coordinate with C atoms to form the Ru-C structures. (iii) Ru-N-Ru species or Ru clusters. According to the guidance of **Reviewer 1#**, the Ru-N-Ru species are also likely to be generated simultaneously during the synthesis of Ru-N-In/NC. The refitted XAS data for the Ru-N-In/NC catalyst further prove this conclusion (**Revised Supplementary Table 4**). Similar results are also reported in several publications.^[R5, R9-R11]

Moreover, despite the presence of various forms of Ru, the Ru-N-In moiety still can be considered as the main active center for acetylene hydrochlorination. Combining catalytic performance testing experiments and characterization data, it can be concluded that the strong stability of Ru-N-In/NC comes from the contribution of Ru-In pairs, not the Ru species without pairing with In atoms. The constructed asymmetric Ru-N-In structure can address the issue of Ru migration induced by the over-chlorination. The introduction of In atoms not only alleviates the steric hindrance, but also enables the selective adsorption of C₂H₂ and HCl on each metal site. Furthermore, a *d-p* orbital coupling between Ru-In pairs modulated the electron configuration of Ru, resulting in the interaction of *p* orbital of Cl with *d*_{x²-y² orbital of Ru. These favorable effects jointly contributed a higher energy barrier for the over-chlorination process, and thermodynamic transition from exothermal to endothermal, which inhibited the migration of Ru atoms, and then dramatically improved the stability for acetylene hydrochlorination. In contrast, the aforementioned interactions between Ru and In atoms are missing on the other Ru species without pairing with In, and the experimental results have demonstrated the presence of excess Ru exerts no significant effect on the stability of Ru-N-In/NC catalysts, which would be removed due to over-chlorination. Optimizing the synthesis methods and increasing the atom pair contents to achieve a better catalytic performance is still a hot and difficult issue, which we will continue to focus and investigate in our future research.}

Figure R3. Structure of Ru-N₄ and Ru-N-In and their free energies calculated by DFT. Orange, pink, blue, and gray spheres represent Ru, In, N, and C atoms, respectively.

Figure R4. AC-HAADF-STEM image of Ru-N-In/NC, in which the signal-atom Ru-In pairs and Ru single atoms are highlighted by yellow ovals and red circles, respectively.

Revised Supplementary Table 4 EXAFS fitting parameters at the Ru and In *K*-edge for Ru-N-In/NC.

Sample	Coordination	CN	R (Å)	σ^2 (Å ²)	ΔE_0 (eV)	R factor
Ru-N-In/NC	Ru-N	3.4±0.4	2.008±0.011	0.0043±0.0019	0.2±1.6	0.0014
	Ru-N-Ru/In	1.7±0.5	3.610±0.017	0.0102±0.0049	-8.5±0.5	
	In-N	4.1±0.7	2.130±0.006	0.0056±0.0016	0.9±0.3	0.0013
	In-N-In/Ru	1.8±0.6	3.608±0.028	0.0132±0.0059	-9.6±2.1	

Note that *CN*: coordination numbers; *R*: bond distance; σ^2 : Debye-Waller factors; ΔE_0 : the inner potential correction; *R* factor: goodness of fit. A reasonable range of EXAFS fitting parameters: $0.700 < S_0^2 < 1.000$; $CN > 0$; $\sigma^2 > 0 \text{ \AA}^2$; $|\Delta E_0| < 10 \text{ eV}$; *R* factor < 0.02 . Fitting range: $3.0 \leq k \text{ (/}\text{\AA}) \leq 13.5$ and $1.0 \leq R \text{ (\AA)} \leq 3.0$ (Ru foil); $3.0 \leq k \text{ (/}\text{\AA}) \leq 10.5$ and $1.0 \leq R \text{ (\AA)} \leq 4.0$ (**Ru-N-In/NC**); $3.0 \leq k \text{ (/}\text{\AA}) \leq 11.0$ and $2.0 \leq R \text{ (\AA)} \leq 4.0$ (In foil); $3.0 \leq k \text{ (/}\text{\AA}) \leq 10.5$ and $1.0 \leq R \text{ (\AA)} \leq 4.0$ (Ru-N-**In/NC**).

[R1] Li, Z. et al. Well-Defined Materials for Heterogeneous Catalysis: From Nanoparticles to Isolated Single-Atom Sites. *Chem. Rev.* **120**, 623-682 (2020).

[R2] Kaiser, S. et al. Performance Descriptors of Nanostructured Metal Catalysts for Acetylene Hydrochlorination. *Nat. Nanotechnol.* **17**, 606-612 (2022).

[R3] Liu, L. and Corma, A. Metal Catalysts for Heterogeneous Catalysis: From Single Atoms to Nanoclusters and Nanoparticles. *Chem. Rev.* **118**, 4981-5079 (2018).

[R4] Martín, A. J. et al. Unifying Views on Catalyst Deactivation. *Nat. Catal.* **5**, 854–866 (2022).

[R5] Li, R. et al. Polystyrene Waste Thermochemical Hydrogenation to Ethylbenzene by a N-Bridged Co, Ni Dual-Atom Catalyst. *J. Am. Chem. Soc.* **145**, 16218-16227 (2023).

[R6] Yang, Q. et al. Understanding the activity of Co-N_{4-x}C_x in atomic metal catalysts for oxygen reduction catalysis. *Angew. Chem. Int. Ed.* **59**, 6122-6127 (2020).

[R7] Gong, Y. N. et al. Regulating the coordination environment of MOF-templated single-atom nickel electrocatalysts for boosting CO₂ reduction. *Angew. Chem. Int. Ed.* **59**,

2705-2709 (2020).

[R8] Pei, J. et al. A replacement strategy for regulating local environment of single-atom Co-S_xN_{4-x} catalysts to facilitate CO₂ electroreduction. *Nat. Commun.* **15**, 416 (2024).

[R9] Liu, C. et al. Catalytic Activity Enhancement on Alcohol Dehydrogenation via Directing Reaction Pathways from Single- to Double-Atom Catalysis. *J. Am. Chem. Soc.* **144**, 4913-4924 (2022).

[R10] Ji, S. et al. Chemical synthesis of single atomic site catalysts. *Chem. Rev.* **120**, 11900-11955 (2020).

[R11] Zhang, W. et al. Emerging dual-atomic-site catalysts for efficient energy catalysis. *Adv. Mater.* 2021, **33**, 210257 (2021).

Reviewer 3#

I want to thank the authors for their improved manuscript and the methodical response to the questions provided by the reviewers. The authors have answered all of them rigorously and without any ambiguity. Regarding the new manuscript, there is still some room for improvement, such as, a better efficiency in the synthesis of Ru-In pairs, the manuscript is complete for publication.

Response: Thank you for your careful review and kind recommendation. Admittedly, as you mentioned, there is still some room that need to be improved in our research, such as a better efficiency in the synthesis of single-atom pairs. This is the direction and goal that our research team has been working on for a long time. In future studies, our research team will further involve in the optimization of the synthesis methods to increase the content of single-atom pairs, thus achieving a better performance in acetylene hydrochlorination.

1. I would like to remember to add error bars to the new graphs, especially the ones with trend lines (for example, Supplementary Figures 35, 36, 51, Figure 4, etc.).

Response: Thanks for your careful comment. We have added error bars in **Figures 4e~f** and **Figures 5e** as well as **Supplementary Figures 31, 35, 36, 38, 51, 66**.

Figure 4e. The adsorption energy of C₂H₂ and HCl versus the TOF of the Ru-N-In/NC, RuN₄/NC, and InN₄/NC catalysts. The error bars indicate the standard deviations of three experimental measurements. **Figure 4f.** d -band center as a descriptor versus the TOF of the Ru-N-In/NC, RuN₄/NC, and InN₄/NC catalysts, in which the projected density of states analysis of Ru-N-In/NC was inserted. The error bars indicate the standard deviations of three experimental measurements.

Figure 5e Coke deposition of the post-hydrochlorination Ru-based catalysts, determined by the weight loss differences. The error bars indicate the standard deviations of three experimental measurements.

Supplementary Figure 31. Coke deposition (%), determined by thermogravimetric analysis (the amount of deposited coke is calculated from the weight loss difference for the fresh and used catalysts), of Ru-N-In/NC treated by different *GHSV*(C₂H₂) conditions. The error bars indicate the standard deviations of three experimental measurements.

Supplementary Figure 35. Apparent activation energies (kJ/mol) of the as-prepared catalysts obtained from Arrhenius equation, in which the concentrations of C₂H₂ and HCl ranged from 20 to 40%. The error bars indicate the standard deviations of three experimental measurements.

Supplementary Figure 36. Reaction orders (s) obtained from kinetic studies for (a) C₂H₂ and (b) HCl over Ru-N-In/NC, RuN₄/NC, and InN₄/NC. The partial reaction order of both reactants is indicated by the slope of the fitting lines. Each point was determined in an independent test to eliminate the interference of catalyst deactivation. The error bars indicate the standard deviations of three experimental measurements. [Reaction conditions: P = ambient pressure, T = 180 °C, HCl/C₂H₂ = 1.15, and the concentrations of C₂H₂ and HCl was 10 to 20%]

Supplementary Figure 38. Comparison of TOF ($\text{mol}_{\text{C}_2\text{H}_2}/\text{mol}_{\text{metal}}/\text{h}$) of Ru-N-In/NC with the AuIn/NC, PdIn/NC, and PtIn/NC catalysts. The obtained data of the three samples are the same as the reaction conditions of Ru-N-In/NC. The error bars indicate the standard deviations of three experimental measurements.

Supplementary Figure 51. The correlation between activity and optimal acetylene interaction for Ru-N-In/NC. The error bars indicate the standard deviations of three experimental measurements.

Supplementary Figure 66. Ru leaching rate of Ru-N-In/NC and RuN₄/NC with increasing temperature from 120 to 300 °C for 12 h under acetylene atmosphere. The error bars indicate the standard deviations of three experimental measurements.

Special thanks to you for your comments and contributions!

Sincerely,

Pengfei Xie

Professor,

College of Chemical and Biological Engineering

Zhejiang University

REVIEWERS' COMMENTS

Reviewer #1 (Remarks to the Author):

Most of the questions were well addressed. It can be accepted. The typing error;"Rn-N-In" should be corrected.

Reviewer #2 (Remarks to the Author):

The authors have addressed all my concerns and this manuscript can be recommended for publication in its current form

Reviewer #3 (Remarks to the Author):

From my side, the revised manuscript answers all my previous questions.

Point by Point Response to Reviewer's Comments

Reviewer 1:

Most of the questions were well addressed. It can be accepted. The typing error;"Rn-N-In" should be corrected.

Response: We are very grateful to the reviewer for his/her helps and positive comments on our manuscript. Also, we have double checked the entire manuscript and then corrected the "Rn-N-In" to "Ru-N-In".

Reviewer 2:

The authors have addressed all my concerns and this manuscript can be recommended for publication in its current form

Response: We are very grateful to the reviewer for his/her helps and positive comments on our manuscript.

Reviewer 3:

From my side, the revised manuscript answers all my previous questions.

Response: We are very grateful to the reviewer for his/her helps and positive comments on our manuscript.